# Spatiotemporal Distributions of Icebergs in a Temperate Fjord: Columbia Fjord, Alaska

Sarah U. Neuhaus[1], Slawek M. Tulaczyk[1], Carolyn Branecky Begeman[1,2]

[1]Earth and Planetary Sciences, University of California Santa Cruz, Santa Cruz, CA, 95060, USA
[2]Los Alamos National Laboratory, Los Alamos, NM, 87545, USA

*Correspondence to*: Sarah U. Neuhaus (suneuhau@ucsc.edu)

**Abstract.** Much of the world's ice enters the ocean via outlet glaciers terminating in fjords. Inside fjords, icebergs may affect glacier-ocean interactions by cooling incoming ocean waters, enhancing vertical mixing, or by providing back stress on the terminus. However, relatively few studies have been performed on iceberg dynamics inside fjords, particularly outside of Greenland. We examine icebergs calved from Columbia Glacier, Alaska, over eight months spanning late winter to mid-fall using 0.5-meter resolution satellite imagery, identifying icebergs based on pixel brightness. Iceberg sizes fit a power-law distribution with an overall power-law exponent, $m$, of -1.26 ± 0.05. Seasonal variations in the power-law exponent indicate that brittle fracture of icebergs is more prevalent in the summer months. Combining our results with those from previous studies of iceberg distributions, we find that iceberg calving rate, rather than water temperature, appears to be the major control on the exponent value. We also analyze icebergs' spatial distribution inside the fjord and find that large icebergs (10,000 m$^2$ – 100,000 m$^2$ cross-sectional area) have low spatial correlation with icebergs of smaller sizes due to their tendency to ground on shallow regions. We estimate the surface area of icebergs in contact with incoming seawater to be 3.0 ± 0.63 x 10$^4$ m$^2$. Given the much larger surface area of the terminus, 9.7 ± 3.7 x 10$^5$ m$^2$, ocean interactions with the terminus may have a larger impact on ocean heat content than interactions with icebergs.

## 1 Introduction

In recent decades, fjord-terminating glaciers have been rapidly losing mass (Larsen et al., 2007; Pritchard et al., 2009), contributing significantly to eustatic sea level rise (Gardner et al., 2013; McNabb and Hock, 2014). High volumes of ice discharge due to iceberg calving and submarine melt have been attributed to contact between the glacier terminus and relatively warm and salty fjord waters (Bartholomaus et al., 2013; Motyka et al., 2003). Current fjord circulation models do not take icebergs into account, though icebergs may modify warm, dense waters entering the fjord by enhancing vertical mixing and by extracting heat through iceberg melt (Carroll et al., 2015; Klinck et al., 1981; Mortensen et al., 2018; Motyka et al., 2003; Rignot et al., 2010). Various studies have examined the iceberg calving process (Bahr, 1995; Chapuis and Tetzlaff, 2014; Hughes, 2002; O'Neel et al., 2003; Warren et al., 2001), as well as the transport and evolution of icebergs in

the open ocean (Bigg et al., 1997; Dowdeswell and Forsberg, 1992; Gladstone et al., 2001; Kubat et al., 2007), but comparably little is known about iceberg evolution inside the fjords where they originate.

Recent studies of icebergs have focused on icebergs calved from Greenland or Antarctic glaciers, however in this study we characterize the size and location of icebergs in a major Alaskan fjord using high-resolution satellite imagery. We examine the differences in iceberg populations over a span of eight months in 2013 to gain insights into their seasonal variability. We also investigate how icebergs evolve along the fjord to better understand where iceberg meltwater is introduced in vertical and horizontal dimensions.

Our analyses focus on the fjord of Columbia Glacier, which connects with Prince William Sound, Alaska. Columbia Glacier is the single largest contributor to ice loss from Alaskan glaciers, accounting for ~6-17% of annual land ice loss from this region (Gardner et al., 2013; Pfeffer, 2015; Rasmussen et al., 2011). Columbia Glacier is also one of the best-studied glaciers in the world. The United States Geologic Survey (USGS) has been instrumenting Columbia Glacier since the 1970's, and the first time-lapse cameras used to study glacier movement and iceberg calving were implemented in 1978 (Meier and Post, 1978; Pfeffer, 2012). From 2012 to 2015, the Prince William Sound Regional Citizen's Advisory Council (PWSRCAC) commissioned a study of Columbia Glacier, which included several field campaigns and geophysical tools, with the aim of better predicting the future behavior of the glacier. PWSRCAC was particularly interested in understanding iceberg discharge, as icebergs that exit Columbia Fjord later intrude on the shipping lanes into and out of the Port of Valdez (Pfeffer, 2012, 2013a, 2013b, 2014a, 2014b, 2015).

Columbia Glacier has a total surface area of around 910 km$^2$ (McNabb et al., 2012a), and is located in central Alaska in the Chugach Mountains (Fig. 1). From 1794 – when the terminus of Columbia Glacier was first mapped by Captain George Vancouver – to 1980, the terminus of the glacier was in a stable location, terminating at the northern end of Heather Island (Meier and Post, 1978; Post, 1975). From 1980 to 2013, the year when the satellite images used in this project were acquired, the glacier retreated approximately 20 km. This retreat revealed a fjord extending north-south, roughly 5 km in width and 20 km in length. At the entrance to the fjord is a submarine end moraine – which shall be referred to as "Heather Moraine" – built by the glacier when it was in its extended Neoglacial position (Meier and Post, 1978). An oceanographic survey of Columbia Fjord completed in 1983 determined that the water column over Heather Moraine was shallow – less than 20 m below Mean Lower Low Water (the average height of the lowest tide over the National Tidal Datum Epoch, as defined by the National Oceanographic and Atmospheric Administration [NOAA]) – and partially exposed (Pfeffer, 2013a; Walters et al., 1988). Pfeffer (2013a) examined more recent bathymetric surveys of Heather Moraine and found that the bathymetry did not change significantly between 1977 and 2005, indicating that very little erosion has occurred. The mean tidal fluctuation in nearby Valdez, as measured by NOAA, is ~3m, with maximum fluctuations up to 5-6m, indicating the maximum water depth above Heather Moraine to be ~25m. Behind Heather Moraine, fjord bathymetry descends to 200 m below sea level. (Walters et al., 1988). Iceberg calving rates increased following the initiation of glacial retreat, and reached a maximum of over 10 km$^3$ yr$^{-1}$ in 1999-2000 (Pfeffer, 2013b) – 8.5 km$^3$ yr$^{-1}$ averaged over 1996-2007 (Rasmussen et al., 2011). Calving rates at Columbia Glacier have since been decreasing (Pfeffer,

2013b).  Between 2010 and 2013 the average ice flux into the fjord was measured at 2.23 km$^3$ yr$^{-1}$ (Pfeffer, 2013b), most of which came from the main branch of the glacier.  Between 2011 and 2014 the average mass flux from the main branch of Columbia Glacier was measured to be 1.18 ± 0.30 Gt yr$^{-1}$ (~1.29 km$^3$ yr$^{-1}$) (Vijay and Braun, 2017).

## 2 Methods

### 2.1 Image Processing

To determine the spatial distribution of icebergs, we obtained 0.5 m-resolution, 8-bit, grayscale imagery of Columbia Fjord from the Polar Geospatial Center at the University of Minnesota.  The images were taken by the WorldView satellites 1 and 2 during 2013, and georeferenced by the Polar Geospatial Center (image details shown in Table 1).  Although the WorldView 2 satellite offered multispectral bands, we only used the higher-resolution panchromatic bands from both satellites (WorldView 1 spectrum spanned wavelengths 400 – 900 nm, and WorldView 2 spectrum spanned wavelengths 450 – 800 nm).  The spread of nine dates provided seasonal coverage of the fjord, from late winter to the middle of the fall: March 13, May 6, June 10, July 11, July 12, and November 19.  Note that we use the oceanographic definitions of seasons, such that winter is January-February-March, and so on in 3-month increments.

To image large swaths, the WorldView satellites capture multiple overlapping images at once.  We mosaiced these overlapping images taken simultaneously or within hundredths of a second to provide complete coverage of the fjord, with the exception of July 11 and May 06b.  For several of the dates a second set of images were taken one minute later, for the purposes of stereo imagery.  We distinguish between the two sets of images by labelling the first image mosaic 'a' and the second 'b'.  Because the icebergs likely remain unchanged between these sets of images, the differences in iceberg identification between 'a' and 'b' images potentially result from changes in reflectivity of the open ocean due to the two distinct satellite viewing angles.  Changing the angle at which we view the ocean changes the reflectivity of the ocean, which could therefore affect which pixels were identified as ice versus water.  We used these pairs of images to constrain the error on our iceberg identification method.

We consider the fjord to be the area bounded by the shores of the fjord, the glacier termini, and Heather Moraine (covering a total area of ~87 km$^2$).  Because of the shallow water depths above Heather Moraine, it is a natural barrier separating Columbia Fjord from Prince William Sound.  In addition, the shallow depths cause large icebergs to run aground, allowing for easy identification of the submarine moraine in the satellite imagery.

To identify and locate icebergs in each image we created a thresholding algorithm using the MATLAB image processing toolbox that identified potential iceberg areas based on pixel intensity value.  Icebergs had higher pixel brightness values than the darker fjord waters, thus we set a threshold brightness value above which pixels were classified as icebergs and below which they were classified as water.  Because frequency histograms of pixel brightness did not reveal a bimodal pattern, we therefore chose a threshold value of 41 out of 255, which corresponded with the highest pixel value for open ocean identified through visual inspection.  The automatic iceberg recognition algorithm performance was impacted when

the icebergs were surrounded by ice mélange, tiny chunks of ice derived from icebergs or the glacier terminus floating on top of the water. Since the mélange was brighter than the water, but darker than the icebergs, we were able to mitigate this by adjusting the threshold value in those areas until we reached a more realistic discrimination of icebergs based on visual inspection. We manually inspected the automatically-selected icebergs to quantify the reliability of automatic iceberg detection. We treat visual inspection as the 'gold standard' because the human observer can use textural and contextual information in addition to brightness alone. However, we opt for an automatic detection of icebergs for this study because mapping of all icebergs by visual inspection would be prohibitively time consuming. An example of iceberg identification by the algorithm is shown in Fig. 2b.

In addition to defining the pixel intensity threshold, we set upper and lower bounds on the iceberg area, with the lower bound corresponding to the smallest group of pixels we could visually identify as an iceberg (20 pixels, or 5 m$^2$) and the upper bound corresponding to the number of pixels in the largest iceberg we found visually (112,000 pixels). We set an upper bound on iceberg size to prevent large areas of ice mélange being identified as a single iceberg. The term 'iceberg' in this study refers to icebergs (>15m across as defined by the Canadian Ice Service, or >3000 ft$^2$ [278 m$^2$] in area as defined by the US National Ice Service), as well as growlers and bergy bits (glacially-derived ice in the ocean that is smaller than an iceberg as typically defined by the previously mentioned Ice Services).

To quantify the error on iceberg identification, we compared the results of the manual and automatic iceberg identification. We divided each mosaic into a test grid, with each test cell being 250 by 250 pixels (~125m x 125m, or roughly half the area of the largest iceberg). We then randomly selected test cells on each of the mosaics and counted the number of icebergs by eye and using the thresholding code. After examining a total of 107 grid cells, we found that the algorithm identified $95 \pm 4$ % of the icebergs identified by an observer. Occasionally, when icebergs were close together, the algorithm would categorize them as a single iceberg, leading to the under-identification of the number of icebergs. To verify if we had counted enough cells, we divided the cumulative number of icebergs identified by the algorithm in cells 1 through N (where N is the number of test cells counted) by the cumulative number of icebergs identified manually (cells 1 through N) (Fig. 2a). This value plateaued for N≥44. We used the mean and the standard deviation of the line between N=44 and N=107 as our estimate of iceberg identification error. Error in automatic iceberg identification was greater in mélange-covered areas than open ocean. We found that the algorithm identified $67 \pm 6$ % of the icebergs in the areas of ice mélange. To assess the amount of mélange present in Columbia Fjord, we examined the pixels identified as ice, but not counted as icebergs. The area of the fjord covered by mélange ranged from $1.1 \pm 0.12$ to $9.9 \pm 0.79$ % (Table 4).

In addition to quantifying the error in the number of icebergs identified, we estimated the error on the area of the icebergs by comparing the total area of icebergs calculated by the algorithm. For June 10a and June 10b, the difference in total iceberg area amounted to less than 8% of the total calculated iceberg area for each image. For July 12a and July 12b, that difference amounted to less than 4%.

## 2.2 Overall Iceberg Distributions

To examine the spatial distribution of icebergs inside the fjord, we created a time-integrated map of iceberg density. We divided the fjord into 500 m x 500 m squares and counted the number of icebergs whose centroids were present within each square. Overlaying the results from March 13, June 10a, July 12a, and November 19, we created a map which depicted the locations of all the icebergs identified during our study (Fig. 4a). We created a similar map of cumulative area of icebergs whose centroids resided in each square (Fig. 4b).

## 2.3 Along Fjord Iceberg Distributions

To examine the variation in spatial distribution of icebergs between scenes, we divided the fjord longitudinally into evenly spaced sections roughly one kilometer in length starting from the terminus to Heather Moraine. The icebergs were binned into those sections based on the location of their centroids. In some of our analyses, we needed a larger sample size of icebergs, so we also divided the fjord into three geographic sections (Fig. 1), which are based on fjord geometry. The "Proximal Zone" spans from the terminus of the glacier to the first major constriction, the "Mid-Fjord Zone" spans from the constriction to the inlet on the east side of the fjord, roughly two-thirds of the way downfjord, and the "Distal Zone" spans from the inlet to Heather Moraine.

Following the methodology established in prior similar studies, we fit a power-law equation of the form $Pr(A > a) = bA^m$ to iceberg areas in each of the three zones to determine the iceberg size-distribution (Kirkham et al., 2017; Sulak et al., 2017; Tournadre et al., 2016). $Pr(A > a)$ represents the number of icebergs whose waterline cross-sectional area, $A$, is greater than $a$ while $b$ and $m$ are the constants which are constrained by data fitting. We chose values of $a$ to be multiples of 1000 from 0 to 100,000 m$^2$, increasing the resolution of $a$ between 0 and 1000 m$^2$ to include: 10, 25, 50, 75, 100, 250, 500, and 750 m$^2$. To determine the goodness of fit of the power-law we performed a Kolmogorov-Smirnov test. To obtain a fit with 95% confidence we had to remove the largest icebergs, which deviate from the power-law size distribution that is followed by smaller icebergs. There is often difficulty fitting the tail of the distribution due to the number of samples being too low, and removing the largest icebergs to fit the distribution has been done in other similar recent studies (e.g. Sulak et al., 2017). An example of our power-law fit for a single image is shown in Fig. 3.

To test whether icebergs of various sizes co-vary spatially within Columbia Fjord, we separated the iceberg population into five class sizes based on their waterline cross-sectional area, which we define as the area of the icebergs visible from satellite imagery. The class size bins were spaced logarithmically (0-10 m$^2$, 10-100 m$^2$, 100-1,000 m$^2$, 1,000-10,000 m$^2$, and 10,000-100,000 m$^2$) with Class 1 representing the smallest icebergs and Class 5 representing the largest. For each one-kilometer bin along the fjord we compared the evolution of the different classes down the fjord by plotting the different class sizes against each other and calculating the Pearson correlation coefficient using the equation:

$$P_{C_a C_b} = \frac{cov(C_a, C_b)}{\sigma_{C_a} \sigma_{C_b}} \qquad \text{Eq. (1)}$$

where $cov$ is the covariance, $\sigma$ is the standard deviation, and $C_a$ and $C_b$ represent the two class sizes being compared. Error bounds are given as 95% confidence intervals. We calculated the Pearson correlation coefficient using the icebergs from all scenes combined. In addition to calculating the correlation of the different size classes, we plotted the exact locations of the large icebergs (Classes 3-5) inside the fjord for each date to better understand where the large icebergs were located inside the fjord (Fig. 8).

## 2.4 Iceberg Characteristics

We used the waterline cross-sectional area of the icebergs ($A$) to calculate iceberg volume. We estimated iceberg volume using two previously proposed scaling laws. For the first approach we used the equation for iceberg volume ($V$) derived by Sulak et al. (2017):

$$V = 6.0A^{1.30} \qquad \text{Eq. (2)}$$

For the second approach we assumed the icebergs to be rectangular prisms and used the proportions used by Bigg et al. (1997) where the ratio of iceberg length to width is equal to 1.5:1, the keel to the width ratio equal to 1:1, and the keel to freeboard ratio equal to 5:1. The equation relating area to volume using these dimensions is:

$$V = 0.98A^{1.5} \qquad \text{Eq. (3)}$$

We also used the dimensions outlined in Bigg et al. (1997) to estimate keel depth ($k$) and underwater surface area (SA) of the icebergs:

$$k = 0.67A^{0.5} \qquad \text{Eq. (4)}$$

$$SA = \frac{11}{3}A \qquad \text{Eq. (5)}$$

We calculated the relative increase in the fjord albedo due to the presence of ice for each scene by assigning an albedo of 0.60 for each pixel identified as ice (Zeng et al., 1984) and 0.060 for the remaining pixels representing ocean surface. In this way, icebergs as well as mélange were accounted for in the ice fraction. The selected albedo of fjord water is the monthly averaged albedo for ocean water surface for the months of April, May, June, August, and September for the latitude of Columbia Fjord (Payne, 1972). We calculated albedo using this method to avoid problems with atmospheric influence on albedo calculations made from satellite imagery as well as to ignore the influence of solar angle on the ocean albedo whilst illustrating the impact of ice in the fjord on fjord albedo.

## 3 Results

### 3.1 Overall Iceberg Distributions

During 2013, the majority of icebergs were found within the first 5 km of the terminus, corresponding to the area of the fjord prior to where the fjord coastline pinches in and forms a constriction (Fig. 4a). Beyond the constriction, the number of icebergs drops steeply (Fig. 4a). Iceberg area followed a similar pattern, with the majority of iceberg area in the first 5 km from the terminus, followed by a rapid decline in total iceberg area on the other side of the constriction (Fig. 4b and Fig. 5).

All scenes show a peak in the total iceberg area not directly adjacent to the terminus, but 2 to 5 kilometers away. Most icebergs were small; over 95% of all the icebergs identified in this study had a waterline cross-sectional area of 100 $m^2$ or less (Classes 1 and 2). The mélange coverage in the spring and fall months was similar ($2.8 \pm 1.8$ x $10^6$ $m^2$), whereas the mélange coverage in the summer months was $9.1 \pm 0.8$ x $10^6$ $m^2$.

## 3.2 Along Fjord Iceberg Distributions

The exponents from the power-law equations fit to the iceberg size distributions reveal patterns in both the spatial and seasonal size distributions of icebergs inside the fjord (Fig. 6). The power exponent represents the relative abundance of large versus small icebergs, with more negative power exponent values indicating a higher proportion of small icebergs. The power exponents for the spring and fall months were similar, whereas the power exponents were more negative for the summer scenes. The power exponent for the spring and fall months ranged from $-1.22 \pm 0.03$ to $-0.91 \pm 0.02$ with a mean of -1.08. In the summer scenes, the power exponent ranged from $-1.54 \pm 0.03$ to $-1.12 \pm 0.07$ with a mean of -1.35. Every scene showed a decrease in the power-law exponent from the Proximal Zone to the Mid-Fjord Zone (indicating a decrease in the proportion of large icebergs with distance from the glacier terminus) and a subsequent increase in the power-law exponent from the Mid-Fjord Zone to the Distal Zone (indicating an increase in the proportion of large icebergs near Heather Moraine). The average decrease in the power exponent from the Proximal-Fjord to the Mid-Fjord Zone was 0.16, and the average increase from the Mid-Fjord Zone to the Distal Zone was 0.24.

We calculated the spatial correlation of the different iceberg size classes along the fjord to reveal similarities and differences in iceberg evolution down the fjord. The correlation coefficient reflects the spatial covariance of different iceberg size classes. We performed these calculations for each scene individually, and for all identified icebergs combined. Our results show that the largest iceberg class, with a waterline cross-sectional area between 10,000 and 100,000 $m^2$, is the least spatially correlated with the other classes (correlation coefficient ranging between $0.344^{+0.131}_{-0.146}$ and $0.490^{+0.111}_{-0.129}$; Table 2). The fact that the largest icebergs behave differently along the fjord than the other icebergs further justifies removing the largest icebergs from the dataset when fitting the power-law equation for size-frequency distribution. In contrast, the other class sizes are highly spatially correlated with each other (correlation coefficients ranging between $0.814^{+0.047}_{-0.061}$ and $0.980^{+0.005}_{-0.007}$; Table 2). After calculating the spatial correlation of the iceberg classes for each scene independently, we determined that the correlation does not appear to be seasonally variable.

## 3.3 Iceberg Volume and Effects on the Fjord

The two approaches we used to calculate iceberg volume yielded slightly different results (see Table 3). Generally, the volume calculations using Eq. (3) were larger than the volume calculations using Eq. (2). Both approaches showed that the icebergs with waterline surface areas greater than 1000 $m^2$ accounted for the majority of the total ice volume present in each scene. Using Eq. (3), the percentage of ice volume that the large icebergs contained ranged from 53% to 88%, and using Eq. (2), the percentage of ice volume that the large icebergs contained ranged from 35% to 74%. The differences in

iceberg volume found by the two methods are due to the differences in iceberg geometry assumed by these equations. Importantly, the variations in total volume and the proportion of iceberg volume in large icebergs are similar for both approaches, despite the different assumptions in iceberg geometry.

The estimated increase in albedo due to the presence of icebergs for Columbia Fjord ranged from 1.2% to 9.8% (Table 4). The albedo increase was highest in the summer months, corresponding to the increased presence of ice inside the fjord.

To evaluate the potential for icebergs to affect fjord waters, we estimated the iceberg residence time inside the fjord. The bulk iceberg residence time for each image is the total volume of icebergs inside the fjord divided by the average annual calving rate for both arms of Columbia Glacier (Pfeffer, 2013b). The average residence time over all images was 14 ± 6 days using Eq. (2) to calculate iceberg volume, and 15 ± 6 days when using Eq. (3).

## 4 Discussion

### 4.1 Overall Iceberg Distributions

Our data reveal spatial patterns in iceberg distribution in Columbia Fjord in 2013. In general, iceberg coverage decreases with distance from the terminus. The observed peak in iceberg ice coverage 2 to 5 km from the terminus (Fig. 5) is somewhat surprising given that icebergs originate at the terminus, and it would be logical to expect the highest concentration of icebergs to be immediately adjacent to the terminus. Potential explanations for this are that the kinetic energy associated with the calving process or that the inflow of subglacial meltwater at the grounding line pushes icebergs away from the terminus. Furthermore, only the summer months showed peak iceberg concentration away from the terminus (normalizing total iceberg area to bin area), pointing to a causal role for subglacial meltwater discharge, which is higher in the summer. Alternatively, these patterns of ice concentration in the fjord could be the result of influx of icebergs from the west arm of Columbia Glacier. Figures 4a and 4b show an increase in both the number and area of icebergs in the location where the west arm of Columbia Glacier contacts the fjord. Additionally, ocean circulation patterns within the fjord could be driving these patterns of ice congregation. Near the terminus the fjord is wide, but roughly 4 to 6 km downfjord it narrows to ~2 km before expanding out to a consistent width of ~4.5 km until Heather Moraine. This change in geometry may drive ocean circulation that concentrate icebergs 2-5 km from the terminus.

In addition to these spatial patterns there were seasonal differences in iceberg coverage in Columbia Fjord in 2013, with more icebergs present during the summer months than the spring or fall (Fig. 5). This is consistent with an increase in calving rates caused by warmer air and water temperatures. Warmer fjord waters may increase the rate of submarine melt, which then increases the iceberg calving rates (Luckman et al., 2015; O'Leary and Christoffersen, 2013). In addition, surface meltwater caused by warmer air temperatures can aid the formation of icebergs by infiltrating and enlarging crevasses at the terminus (Van Der Veen, 1998; Weertman, 1973). These processes may all work together to produce increased ice discharge during warm summer months as opposed to the late winter and fall. In calculating the calving rate of

Columbia Glacier in 2013, Vijay and Braun (2017) show an increase in the calving rate from March until June/July, followed by a decrease in the calving rate for the remainder of the year. This increased ice discharge would explain increased iceberg coverage during the summer.

## 4.2 Along Fjord Iceberg Distributions

We fit power-law distributions to the data to gain insight – both seasonally and spatially – into the size-distributions of icebergs inside the fjord. Fitting a power-law distribution to the data allows us to more quantitatively understand the spatiotemporal differences in iceberg size distribution because power-law exponents reflect the relative abundance of small icebergs versus large icebergs. Our iceberg distributions were better described by a power-law distribution than by a lognormal distribution, which is consistent with the conclusion from Kirkham et al. (2017) that icebergs near the calving

front tend toward a power-law distribution, and icebergs further out in the open ocean fit a lognormal distribution. The decrease in power-law exponents from the Proximal Zone to the Mid-Fjord Zone indicates a decrease in the relative proportion of large icebergs from the Mid-Fjord Zone to the Proximal Zone. This is unsurprising given that we would expect icebergs to melt or fracture rather than grow as they travel from the terminus (Fig. 6). The proportion of large icebergs increases in the Distal Zone (shown by the increase in power-law exponents between the Mid-Fjord and Distal

Zones), which is interpreted to be due to grounding of icebergs on Heather Moraine, which is at most 25 m below the sea surface (Pfeffer, 2013a; Walters et al., 1988). The larger icebergs become grounded until they have melted or broken up sufficiently to pass over, are pushed over by strong winds, or are able to pass during high tides (Pfeffer, 2015).

Power-law exponents also indicate that there is a greater proportion of smaller icebergs present throughout the fjord in the summer months, including within the Proximal Zone near the terminus. This could indicate that the glacier calves

smaller icebergs in the summer. During the summer when air temperatures are higher, meltwater is ubiquitous along the surface of the glacier and can help break calving ice into smaller pieces through hydrofracturing (Van Der Veen, 1998). Alternatively, the icebergs could be more prone to melt and fragmentation during the summer, which would lead to the increase of small icebergs in the Proximal Zone. The power-law exponents found in the summer months were very close to -1.5, which has been shown both experimentally and theoretically to be indicative of dominant brittle fragmentation (Åström,

2006; Spahn et al., 2014). Previous studies examining iceberg size distributions resulting from fragmentation have also calculated power-law exponents close to -1.5 (Bouhier et al., 2018; Crawford et al., 2018; Tournadre et al., 2016). The warmer summer conditions in Columbia Fjord could be responsible for the increased iceberg fragmentation. For all the environmental parameters examined in their study, Bouhier et al. (2018) found that sea surface temperature was the most highly correlated with iceberg fragmentation rates.

We compared the power-law exponents from Columbia Glacier to those calculated for other glaciers to determine the factors that influence iceberg size distribution. Sulak et al. (2017) reported a power-law exponent of -2.00 ±0.06 for Sermilik Fjord, a power-law exponent of -1.87 ±0.05 for Rink Isbrae Fjord, and a power-law exponent of -1.62 ± 0.04 for Kangerlussuup Sermia Fjord. Kirkham et al. (2017) reported a power-law exponent of -2.4 for the icebergs near the calving

front of Jakobshavn Isbrae, Ilulissat Icefjord. These studies found their power exponents using the icebergs in the entire fjord. Therefore, we also calculated the power exponent for the entire Columbia Fjord, and averaged our results from each image to produce a value of -1.26 ± 0.05 for the entire study. Our results combined with results from these previous studies show no discernible relationship between power-law exponents and seawater temperatures; however, there is a relationship

between power-law exponents and average annual calving flux (Fig. 7) (Howat et al., 2011; Sulak et al., 2017; Vijay and Braun, 2017). A higher calving flux corresponded to a more negative power-law exponent; glaciers with higher discharge rates had higher proportions of small icebergs. This supports the hypothesis of Sulak et al. (2017) that the power-law exponent could be an indicator of glacier productivity, i.e. calving rate.

        In calculating the spatial correlation of icebergs inside the fjord, we found that the majority of icebergs followed similar spatial patterns, but that the largest iceberg class was not strongly spatially correlated to any of the other size classes.

The largest icebergs contain the majority of the ice inside the fjord, yet they behave differently than the remainder of the iceberg population. Our interpretation of this lack of correlation is that the largest icebergs are running aground on the shallower areas of the fjord, which decouples their spatiotemporal evolution from the smaller icebergs that tend to float more freely and evolve together as they move downfjord. Bathymetric surveys (McNabb et al., 2012a) show that in addition to being shallow along the sides and at Heather Moraine, the fjord contains areas near the glacier terminus with depths around

75 m which are able to ground icebergs with waterline cross-sectional areas larger than roughly 8,500 m$^2$ [Eq. (4)]. We found that the largest icebergs were located in those shallower areas (Fig. 8).

### 4.3 Iceberg Effects on the Fjord

        Our findings of iceberg distribution throughout the fjord have direct implications for the locations of freshwater

input. In contrast to riverine fluxes, freshwater fluxes from icebergs can be spatially distributed throughout the fjord; in contrast to precipitation, these fluxes may be spatially concentrated by factors such as wind stress, ocean currents and bathymetry (Bigg et al., 1997). These factors are reflected in the cumulative distribution of icebergs shown in Fig. 4. The sinuosity of the iceberg density in the Distal Zone is likely related to wind stress or ocean current patterns as there are no related bathymetric features. As melt rate is related to the velocity differential between iceberg velocity and ocean current

velocity, either high wind stress or ocean currents could produce velocity differentials that increase melt rates (Bigg et al., 1997). Thus, the sinuous iceberg density feature may be associated with elevated freshwater fluxes relative to other regions of the fjord.

        In contrast, the presence of large icebergs along the coasts and at Heather Moraine (Fig. 8) is best explained by their grounding on bathymetric features. While a majority of the icebergs examined in this study were located within the first five

kilometers of the fjord (Fig. 4b), the largest icebergs also tended to be grounded in the shallow areas of the fjord, namely along the coasts and at Heather Moraine (Fig. 8). Where large icebergs run aground, they release freshwater by melting until they shrink and unground by melting or fracture. Thus, unless fracture processes are dominant, large icebergs may release significant volumes of freshwater over small areas of the fjord. Large icebergs – icebergs with a waterline cross-sectional

area greater than 1000 m$^2$ (Classes 4 and 5) – accounted for less than one percent of the number of icebergs present in the fjord, yet they made up 53-88% of the total iceberg volume in the fjord [Eq. (3)], which is a reflection of the power-law distribution. Freshwater fluxes from these icebergs can have implications for ocean mixing, fjord circulation, and the local ecology (Helly et al., 2011).

5   To assess the icebergs' impact on fjord water mass characteristics, we compared temperature and salinity profiles – collected July 30-31, 2013 (Arimitsu et al., 2017) – to iceberg keel depth (Fig. 9). These profiles show a vertical structure characterized by a diurnally heated surface layer (ca. 1-13 °C), a mixed layer (ca. 4 °C and 26-30 PSU), and a deeper layer (ca. 5 °C and 30 PSU). The salinity of the surface layer is highly variable (ca. 20-26 PSU). The mixed layer is cooler and fresher than the deeper layer and extends to a depth of ca. 30m in the Mid-Fjord and Distal Zones and ca. 60m in the

10 Proximal Zone. Iceberg keel depths are generally coincident with the depth of the mixed layer; 99% of all iceberg keel depths are found within the mixed layer. While it may be a coincidence that the iceberg keel depth is related to the mixed layer depth, there are a few possible causal relationships to consider. The inverse relationship between salinity and number of iceberg keels present could be the result of freshwater from iceberg melting. This freshwater provides a buoyant flux that may significantly enhance vertical mixing from the depth of the iceberg toward the surface (Helly et al., 2011).

15 Additionally, icebergs could mechanically mix the water column by shear produced during iceberg overturning and by current drag against the iceberg surfaces. This could be a significant process at Heather Moraine, where grounded icebergs alter the flow of water in and out of the fjord. Further evidence for iceberg influence on fjord waters at Heather Moraine is that the salinity profiles taken just outside Heather Moraine are ca. 5 PSU higher than directly inside the fjord (Arimitsu et al., 2017).

20   To further assess the icebergs' effect on fjord characteristics, we consider iceberg melt. However, because our images are taken too far apart to track individual icebergs, we cannot directly measure iceberg deterioration. In addition, we cannot calculate iceberg melt rates for all dates using previously published equations due to lack of information of iceberg velocity and fjord water velocity. We therefore estimate an "effective iceberg melt rate" by dividing contemporaneous calving rate calculated in Vijay and Braun (2017) by the underwater surface area calculated using Eq. (5). The effective

25 iceberg melt rate simply represents the rate of iceberg melting that would be required to balance the calving rate given the estimated underwater surface area of the icebergs. Because we lack information about iceberg surface roughness, we neglect it with the consequence that our surface area calculations are an underestimate, and hence, our calculations of iceberg melt rates are an overestimate. Surprisingly, the effective iceberg melt rates were lowest in the summer months ($0.30 \pm 0.02$ m day$^{-1}$) and highest in the spring ($0.84 \pm 0.11$ m day$^{-1}$).

30   To examine what factors might be responsible for this unexpected result, we used the equation for turbulent iceberg basal melting ($M_b$) presented in Bigg et al. (1997):

$$M_b = 0.58 \, \Delta V^{0.8} \frac{\Delta T}{L^{0.2}} \qquad\qquad\qquad \text{Eq. (6)}$$

Where $\Delta V$ is the difference between iceberg velocity and fjord water velocity, $\Delta T$ is the temperature difference between the surface ice temperature – assumed here to be the melting point of ice – and the temperature of fjord water, and $L$ is the along-fjord length of the iceberg. We examined both the median and maximum iceberg length for each date to evaluate the influence of $L$, but the differences in the iceberg melt rates were negligible because $L$ is raised to the power of 0.2. Normalizing $\Delta V,$ $\Delta T,$ and $L$ by the variables in the July 12a scene (the date closest to the date on which the CTD measurements were taken in the fjord) allows us to examine the relative importance of temperature or velocity differential on iceberg melt rates. If we make the end-member assumption that $\Delta V$ remains constant throughout the year and use the observed July 30 seawater temperature of the mixed layer (3 ºC), (Fig. 9), it follows that water temperatures in the spring would have to be higher (8.0 ± 1.1 ºC) than in summer (2.8 ± 0.19 ºC), which is contrary to what is expected. Given this analysis, we attribute the spring increase in iceberg melt rate mainly to increased current shear between icebergs and the surrounding waters ($\Delta V$), which is consistent with previous studies which found higher iceberg melt rates in Greenland fjords in winter due to increased shear (Moon et al., 2018a). $\Delta V$ was 3.4 ± 0.56 m s$^{-1}$ for the spring scenes, and 0.92 ± 0.08 m s$^{-1}$ for the summer scenes. Possible reasons for this increased winter shear could include stronger winter currents and increased iceberg grounding events due to the greater proportion of large icebergs in the winter months. The difference in temperature between icebergs and the surrounding water may be a secondary factor in enhancing melt rates in the spring, however. The temperature that matters most for iceberg melt is the temperature of the water directly adjacent to the icebergs. Most of the icebergs in this study reside in the Proximal Zone, and in the summer this water is possibly cooler due to increased subglacial discharge and runoff in the summer, as well as increased albedo (Table 4), leading to lower iceberg melt rates.

Finally, icebergs can affect fjord water temperatures by altering its surface reflectivity. The presence of icebergs lowers the amount of solar radiation absorbed by the fjord by increasing the overall albedo of the fjord by 1.2 ± 0.46 to 9.8 ± 3.7 %. Although this increase in albedo is small, it is mostly concentrated near the terminus because the majority of icebergs are found within five kilometers of the terminus. This may be reflected in Fig. 9, which shows that only the surface waters in the Proximal Zone are not subject to warming from insolation, perhaps partly due to the high concentration of reflective icebergs there.

## 4.4 Comparison to Greenland Fjords

Most recent investigations of icebergs in fjords have focused on the peripheries of the Greenland ice sheet (Enderlin et al., 2016; Enderlin and Hamilton, 2014; Kirkham et al., 2017; Moon et al., 2018b; Sulak et al., 2017), which share some similarities to and some differences from Columbia Fjord. One difference between our study site and a number of the sites in Greenland is the presence of winter sea ice. Sea ice formation helps create a thick mélange by preventing icebergs and bergy bits from exiting the fjord. This mélange not only increases iceberg residence time in the fjord, but also provides a back stress on the terminus that slows the rate of iceberg calving and terminus velocity (Amundson et al., 2010; Walter et al., 2012). In Greenland, winter sea ice formation is widely prevalent (Amundson et al., 2010; Higgins, 1991; Walter et al.,

2012), however, sea ice was not present in any of the scenes we examined in detail. Some pancake ice was visible in a satellite image taken on March 26 which was not used in this study, however it was not thick enough to lock in icebergs or provide significant backstress on the glacier terminus. Because maximum sea ice extent in the Arctic is typically reached in March, and oceanographic definition of winter is January-February-March, we consider the March 13 scene to be

representative of winter conditions in Columbia Fjord. We found that the mélange coverage was greatest in the summer months when the iceberg coverage was also greatest, however the mélange coverage only amounted to $11 \pm 0.99$ % (Table 4). Hence, ice mélange in Columbia Fjord appears to be more a function of higher summer calving rates and ice fragmentation rather than resulting from winter-time capture of ice fragments in sea ice. The lack of ice mélange in Columbia Fjord may contribute to the relatively short residence time (a fortnight) of icebergs in this fjord compared to

iceberg residence times of over 100 days in some Greenlandic fjords (Sulak et al., 2017; Sutherland et al., 2014).

Icebergs in Greenlandic fjords are often much larger than the ones observed in Columbia Fjord. The largest icebergs in our study have a waterline cross sectional area on the order of magnitude $10^4\,\mathrm{m}^2$, whereas other studies have measured iceberg areas to be around $10^7\,\mathrm{m}^2$ (Kirkham et al., 2017; Sulak et al., 2017). A primary reason for the smaller icebergs in Columbia Fjord is the small height of the calving front. Vijay and Braun (2017) measured the maximum

thickness of the terminus to be ca. 300 m between July 2011 and July 2014. One consequence of the small icebergs is the reduced presence of icebergs that penetrate the deeper, incoming water layer.

The iceberg surface area in contact with the incoming waters (averaging $2.8 \pm 0.58$ x $10^4$ m$^2$ over all images) is a small fraction of the surface area of the terminus ($9.7 \pm 3.7$ x $10^5$ m$^2$) calculated using the ice thickness data published by McNabb et al. (2012b). This is a conservative estimate of terminus area because we do not account for the sinuosity or

roughness of the terminus, which we also neglected when calculating iceberg surface area. Because we estimate the surface area of the terminus to be approximately two orders of magnitude greater than the surface area of icebergs in contact with the incoming water, we do not believe that the icebergs are contributing as much freshwater to the fjord waters as the terminus itself. However, this should not diminish the importance of icebergs' impact on fjord circulation. We see from the temperature and salinity profiles (Fig. 9) that icebergs alter the water masses in the fjord system, introducing melt water and

forcing mixing of the water column at Heather Moraine. But we do expect iceberg contributions to fjord dynamics to be more significant in fjords where the surface area of icebergs is much greater than the surface area of the glacier terminus.

To highlight the similarities and differences between Columbia Fjord and fjords which have been studied in Greenland, we compare temperature profiles from Columbia Fjord and Sermilik Fjord. The July temperature profiles from Columbia Fjord look similar to summer temperature profiles collected in Sermilik Fjord (Arimitsu et al., 2017; Moon et al.,

2018b). Below 200 m depth in both fjords, the water temperature is ca. 4 ºC, but the temperature of the waters above that depth are a few degrees warmer in Columbia Fjord. This is significant because, unlike in Sermilik Fjord where icebergs are large enough to reach 200 m depth, the icebergs in Columbia Fjord are not large enough to reach the lower warmer layer. Another difference in the temperature profiles between these two fjords is the temperature of the surface waters. The surface temperatures in Columbia Fjord reach as high as 13 ºC, whereas the summer surface temperatures in Sermilik Fjord are close

to freezing. Ice mélange and iceberg albedo likely play some role in this difference, but it is beyond the scope of this study to quantify these effects. These warmer temperatures in Columbia Fjord could be accelerating iceberg melt and fragmentation. While we do not have velocity measurements for the fjord waters nor the icebergs in Columbia Fjord, our analysis of the effective melt rates we calculated suggest that either the velocity of the water or the velocity of the icebergs

increases in the winter months compared to the summer months, resulting in greater shear between icebergs and fjord water.

## 5 Conclusion

In this study we have obtained constraints on the distributions of icebergs inside a large Alaskan temperate fjord with high calving fluxes. Most icebergs were found within five kilometers of the terminus, but peak iceberg frequency was reached a few kilometers away from the terminus, particularly in the summer. The iceberg distributions fit a power-law

distribution as opposed to a lognormal distribution. The power-law exponents suggest that the icebergs melt or break up as they move away from the terminus, and that large icebergs run aground on Heather Moraine. More icebergs were present in the summer months, but those icebergs tended to be smaller. Because the power-law exponents for the summer images are closer to -1.5 than the exponents for the fall and spring images, we attribute the summer increase in small icebergs to intensified iceberg fragmentation by warmer fjord conditions. In addition, we find a correlation between power-law

exponents and average annual calving rate, with larger calving rates resulting in increased proportions of small icebergs.

Most of the calved ice was contained within only a small fraction of large icebergs. The largest icebergs (which account for the majority of calved ice) are the least spatially correlated with the other iceberg class sizes, which we attribute to their tendency to ground in shallow areas of the fjord – namely along the coast and on Heather Moraine. The largest icebergs have the greatest potential to cool the incoming ocean waters before they reach the terminus, however, in Columbia

Fjord the surface area of the glacier terminus is thought to surpass the surface area of icebergs in contact with the incoming ocean water, rendering the iceberg cooling effect somewhat less important. The total surface area of the icebergs inside the fjord amounts to 2.9% of the terminus surface area. We expect that only the icebergs at Heather Moraine have the potential to affect the dynamics of the fjord since the shallow water column allows more contact between icebergs and the incoming water. The icebergs do have the potential to cool the outgoing upper layer of ocean waters by increasing the albedo in the

summer months, and thereby decrease the solar heating.

Icebergs can affect fjord circulation through spatially distributed introduction of meltwater that is dependent on wind stress, fjord currents, and bathymetry. Ocean and wind forcings control where smaller icebergs release meltwater into the fjord, whereas fjord bathymetry controls where the largest icebergs release meltwater. Salinity and temperature profiles indicate that icebergs may influence the mixed layer depth. To examine further the influence of icebergs on fjord freshwater

budget, we calculated an effective melt rate, and found that melt rates were surprisingly higher in the spring months. We speculate that this higher melt rate in the spring months is due to increased shear between icebergs and fjord water rather than increased water temperatures. Freshwater input from icebergs is typically omitted from fjord circulation models. By

helping determine the relative importance of the variables affecting the location and quantity of iceberg melt, this study informs models of glacier-ocean interactions.

## Data Availability

The imagery used in this study are available via the Polar Geospatial Center.

## Author Contributions

SN and ST conceived of the study presented here. SN acquired the data and performed analysis and interpretation of the data with guidance from ST and CB. SN prepared the manuscript with contributions from both ST and CB.

## Competing Interests

The authors declare no conflict of interest.

## Acknowledgements

This material is based upon work supported by the National Aeronautics and Space Agency grant NNX08AD31G. Geospatial support for this work provided by the Polar Geospatial Center under NSF-OPP awards 1043681 and 1559691. We would like to acknowledge Shad O'Neel for inspiration and discussion of this study. Two anonymous reviewers provided helpful feedback on an earlier version of this paper.

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

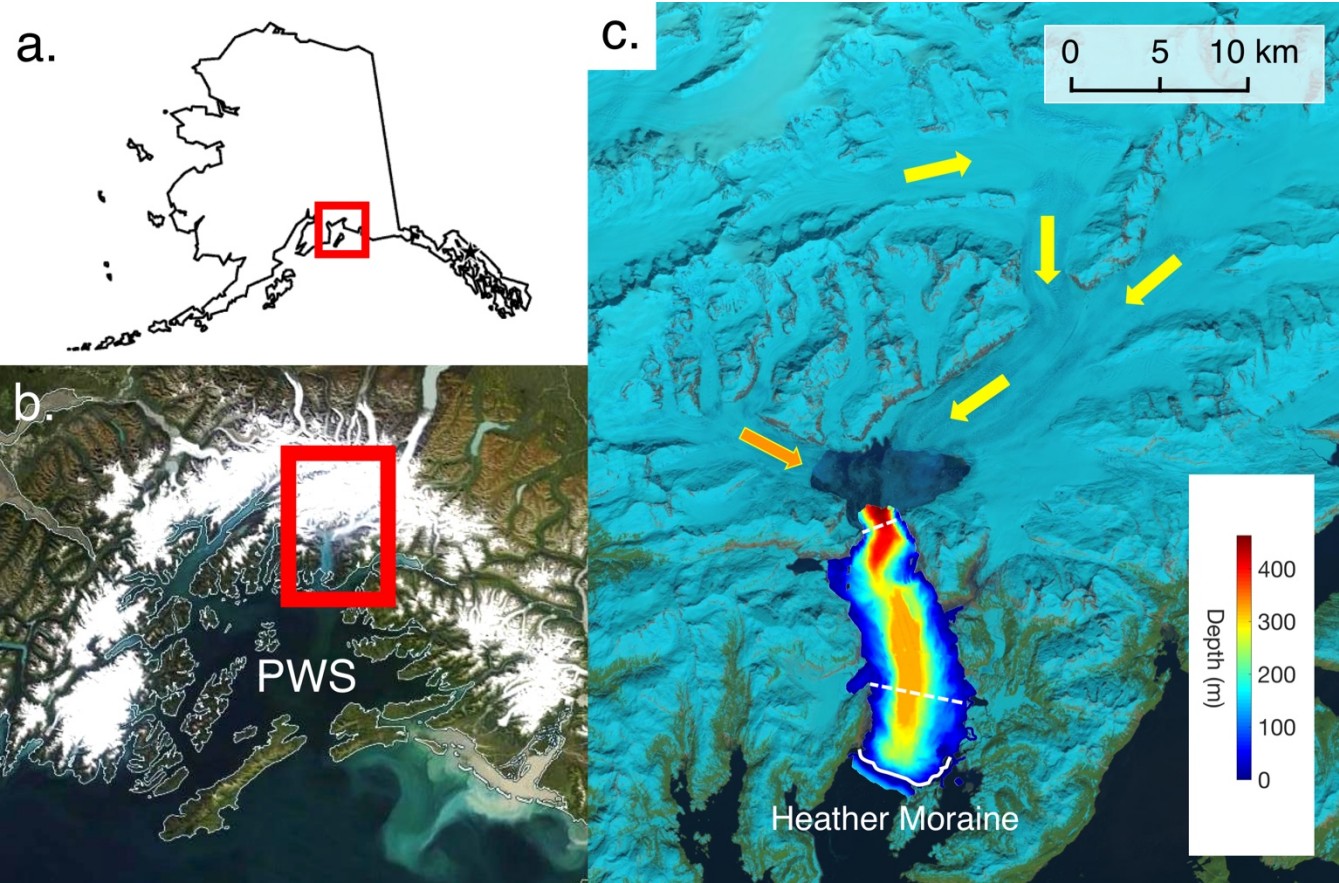

**Figure 1**: Columbia Glacier. **(a)** Outline of Alaska. **(b)** NASA MODIS image of Prince William Sound (PWS) from WorldView satellite. **(c)** Landsat image of Columbia Glacier and Fjord in 2013. Yellow arrows indicate flow of the main branch of the glacier. Orange arrow
15   indicates flow of west branch of the glacier. Heather Island is visible along Heather Moraine. White line indicates location of Heather Moraine. Dashed lines delineate the boundaries between the Proximal Zone, the Mid-Fjord Zone, and the Distal Zone. The bathymetry, measured by NOAA Ship RAINIER in 2005, is overlain on the lower portion of Columbia Fjord.

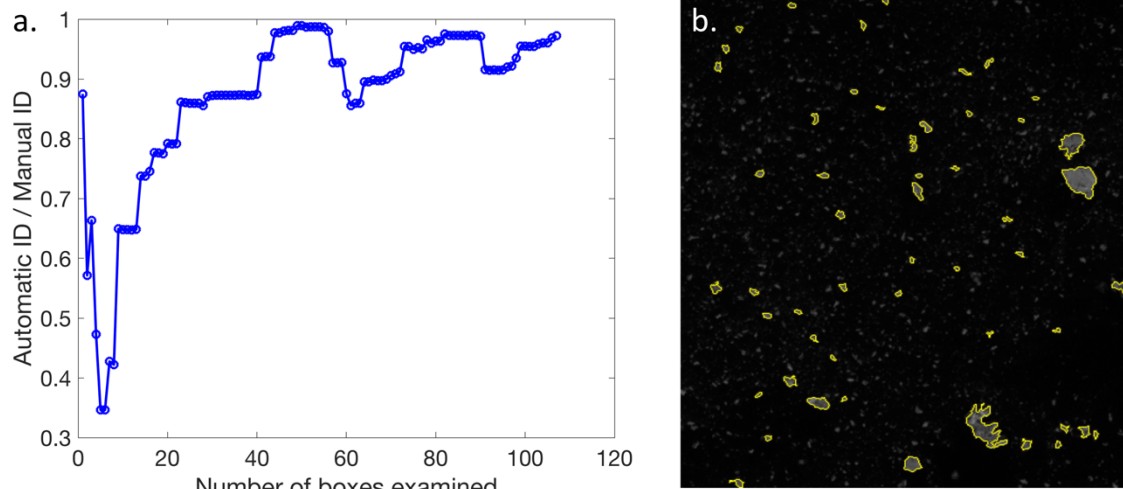

**Figure 2: (a)** The ratio of automatic iceberg detections to manual iceberg detections as a function of the cumulative number of test boxes counted (each box is 250 x 250 pixels). The total number of test boxes counted was 107. **(b)** Example of iceberg detection by algorithm. Gray 'blotches' not outlined in yellow are too small to be identified as icebergs, and are thus classified as mélange.

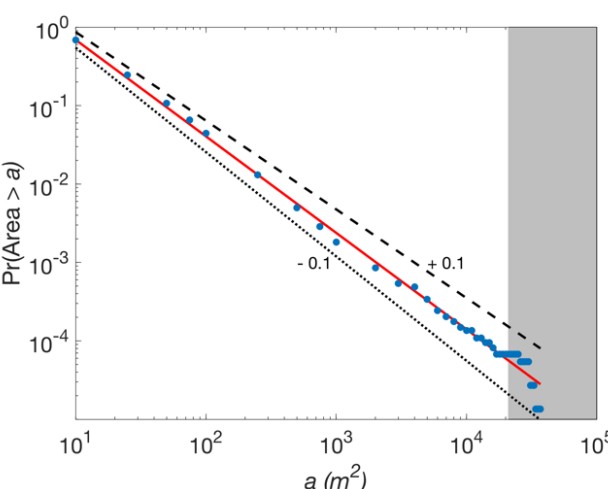

**Figure 3:** Power-law fit for the iceberg areas in the Proximal Zone on June 10b. The red line represents the best fit, and the black lines show the mismatch resulting from shifting the power exponent by ± 0.1 (the dotted line indicates a subtraction of 0.1, and the dashed line indicates an addition of 0.1). The largest icebergs that have been omitted to achieve a statistically significant fit are plotted in the gray

10    box.

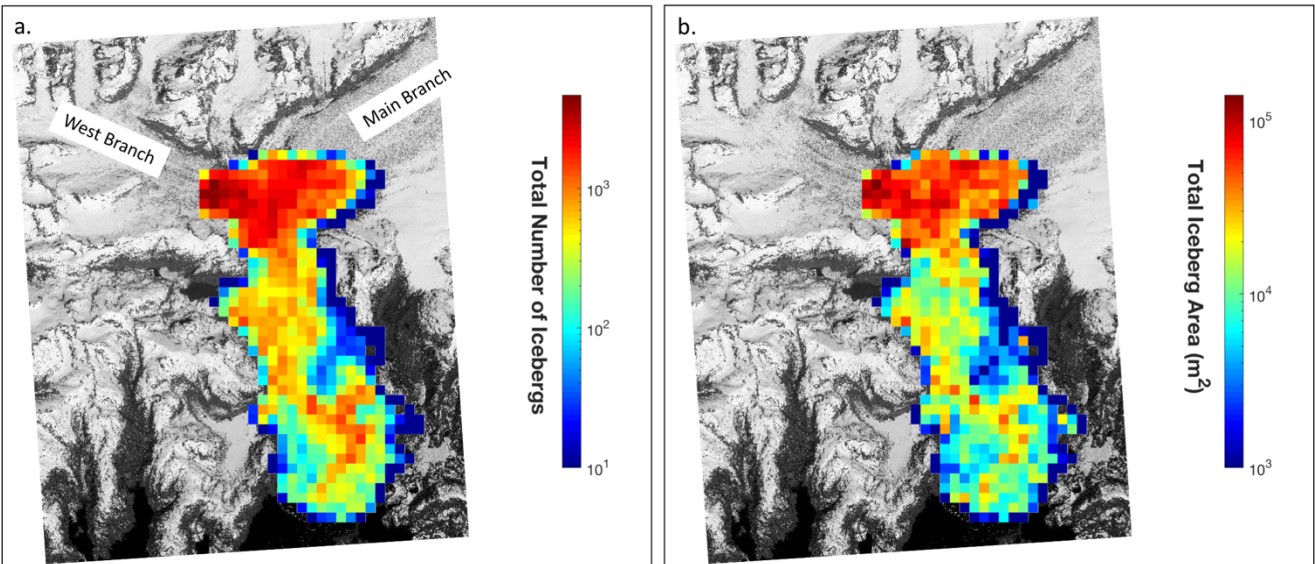

**Figure 4:** Cumulative iceberg population density and area map. Each grid cell represents a 500m x 500m square. The scenes from March 13, May 06a, June 10a, July 12a, and November 19 were overlain to obtain the total number and area of icebergs inside each grid cell. Data is overlain on a satellite image of the fjord taken by WorldView 1 on June 10, 2013. **(a)** number of icebergs in each grid cell **(b)** area of icebergs inside each grid cell.

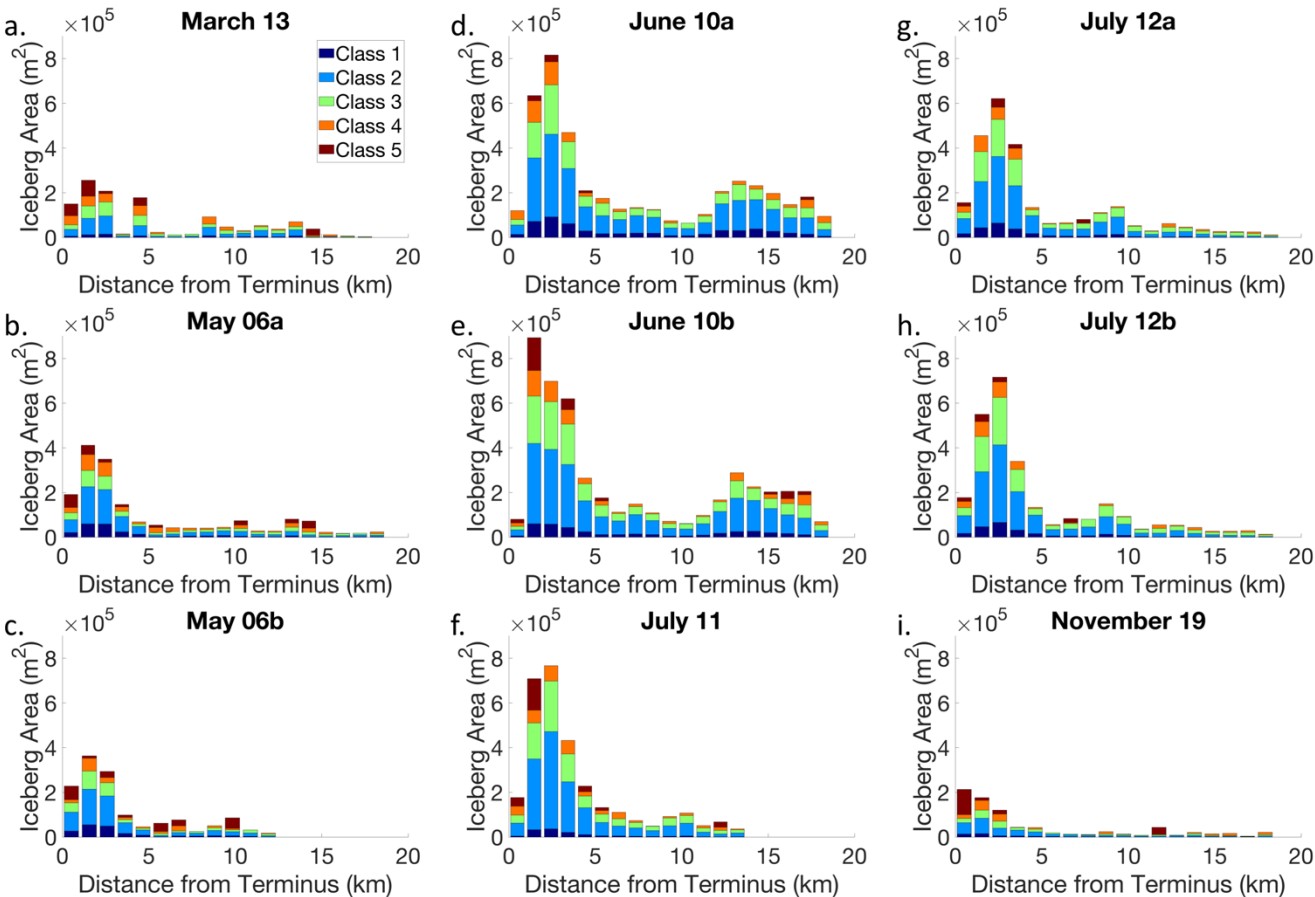

**Figure 5:** Total area of icebergs per 1-km bin along the fjord. The dates shown here are **(a)** March 13, **(b)** May 06a, **(c)** May 06b, **(d)** June 10a, **(e)** June 10b, **(f)** July 11, **(g)** July 12a, **(h)** July 12b, and **(i)** November 19. The peak of ice coverage inside the fjord is found 2-3 km from the terminus. The icebergs are divided into classes by waterline cross-sectional area, with the smallest icebergs residing in Class 1 and the largest in Class 5. The divisions for the bins were: 0-10 $m^2$, 10-100 $m^2$, 100-1,000 $m^2$, 1,000-10,000 $m^2$, and 10,000-100,000 $m^2$.

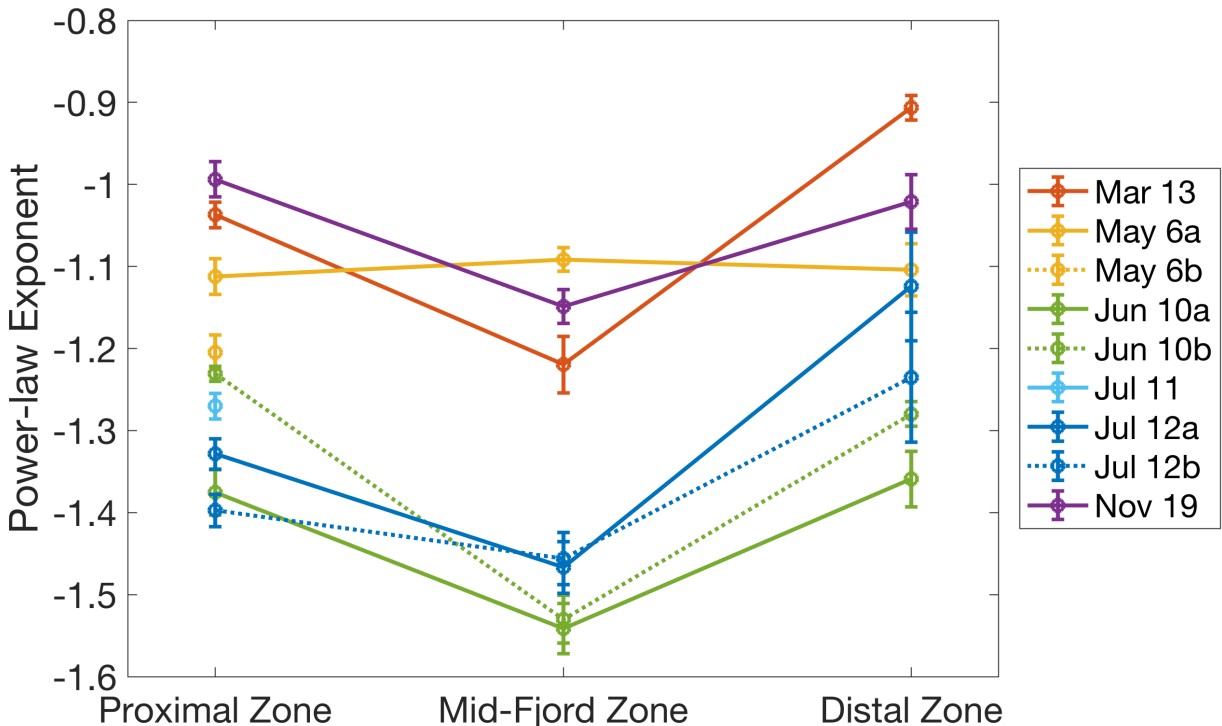

**Figure 6:** Power-law exponent calculated for each scene. Power-law exponents indicate both a spatial and seasonal changes in iceberg distribution. The more positive exponent in the Proximal and Distal Zones indicate a higher proportion of large icebergs present near the terminus and Heather Moraine. Additionally, the more positive exponent for the spring and fall scenes indicates a higher proportion of large icebergs present in those respective seasons. The anomalous increase in the power exponent in the mid-fjord zone for May 6a is due to contamination from cloud cover.

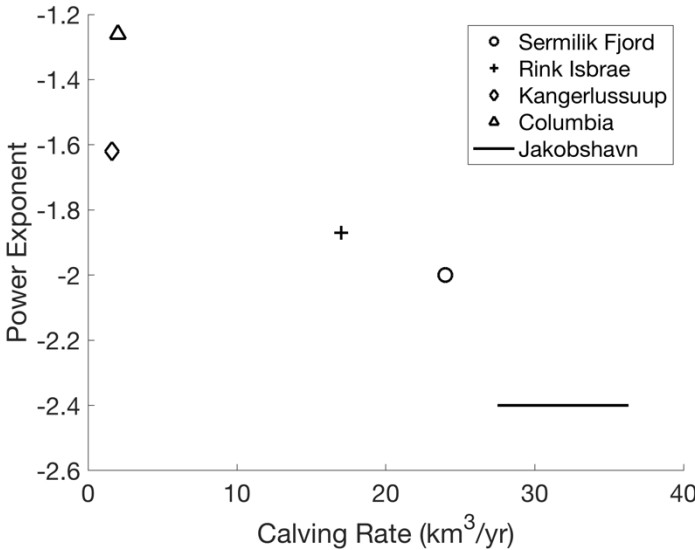

**Figure 7:** A comparison of published power exponents and glacier calving rates. (The power-law exponent for Columbia Fjord was calculated in this study.) An increase in the calving rate corresponds with an increase in the proportion of small icebergs present inside the fjord (more negative power exponent).

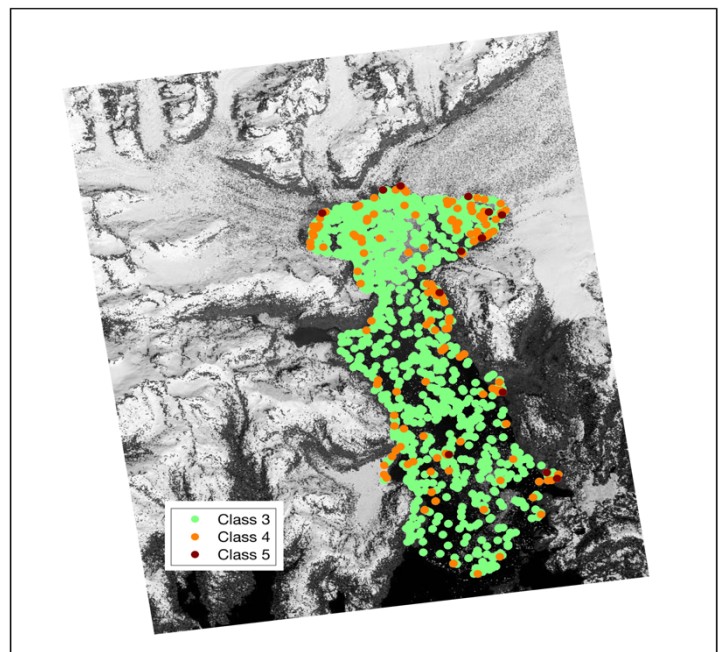

**Figure 8:** Location of large icebergs by centroid (May 06a). Icebergs in Class 3, Class 4, and Class 5 have waterline cross-sectional areas of $100 - 1,000$ m$^2$, $1,000 - 10,000$ m$^2$, and $10,000 - 100,000$ m$^2$ respectively. The background satellite image was taken by WorldView 1 on June 10, 2013.

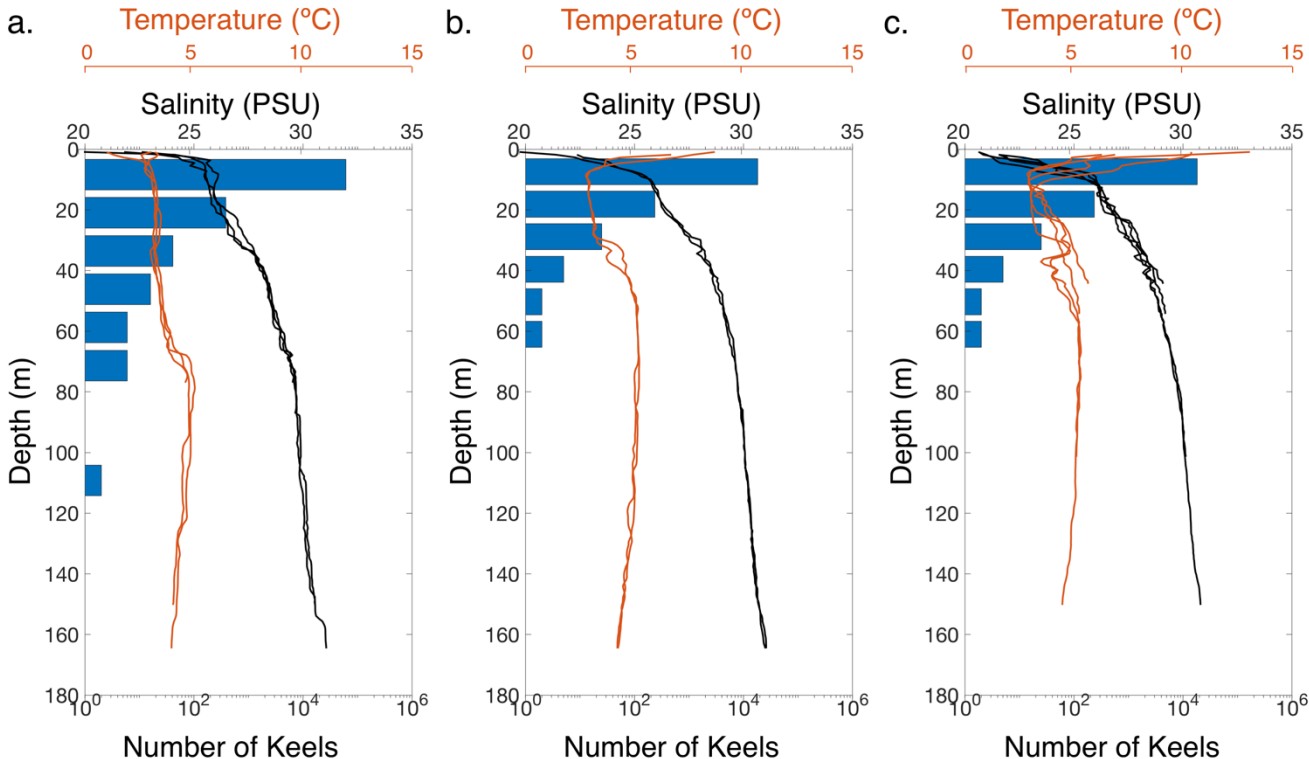

**Figure 9:** Iceberg keel depths compared to salinity and temperature profiles in Columbia Fjord. Salinity and temperature profiles were collected by the US Geological Survey on July 30 and 31, 2013. Keel depths presented here are from the July 12a icebergs. (a) Proximal Zone (b) Mid-Fjord Zone (c) Distal Zone. Note the log axis for number of keel depths.

Table 1: Survey of images used in this study. Image processing was performed by the Polar Geospatial Center prior to our obtaining the images. Satellites used were WorldView 1 and 2 (WV1, WV2). Projection is NAD83. *Time reported in AKDT despite daylight savings ending on November 3.

| Reference ID | Date, 2013 | Alaska Daylight Time | Sensor | Coverage | Number of Images in Mosaic |
|---|---|---|---|---|---|
| March 13 | March 13 | 13:14:44 | WV1 | Full | 2 |
| May 06a | May 06 | 13:32:05 | WV2 | Full | 3 |
| May 06b | May 06 | 13:33:02 | WV2 | Partial | 2 |
| June 10a | June 10 | 13:20:07 | WV1 | Full | 3 |
| June 10b | June 10 | 13:20:52 | WV1 | Full | 3 |
| July 11 | July 11 | 12:51:12 | WV1 | Partial | 3 |
| July 12a | July 12 | 14:03:23 | WV2 | Full | 3 |
| July 12b | July 12 | 14:04:23 | WV2 | Full | 3 |
| November 19 | November 19 | 13:07:18* | WV1 | Full | 2 |

Table 2: Correlation between different iceberg class sizes along the fjord for all scenes combined. The red shading corresponds to the size of the error estimates, with the darker shades of red representing larger error. Upper and lower limits on the 95% confidence interval are reported in the lower half of the table.

| Size Classes | 1 | 2 | 3 | 4 | 5 |
|:---:|:---:|:---:|:---:|:---:|:---:|
| **1** |  | 0.900 | 0.861 | 0.814 | 0.344 |
| **2** | + 0.026<br>- 0.035 |  | 0.980 | 0.845 | 0.398 |
| **3** | + 0.036<br>- 0.047 | + 0.005<br>- 0.007 |  | 0.867 | 0.396 |
| **4** | + 0.047<br>- 0.061 | + 0.040<br>- 0.052 | + 0.034<br>- 0.045 |  | 0.490 |
| **5** | + 0.131<br>- 0.146 | + 0.124<br>- 0.141 | + 0.125<br>- 0.141 | + 0.111<br>- 0.129 |  |

Table 3: Volume of icebergs in Columbia Fjord for each scene, calculated using Eq. (2) and Eq. (3). * Imagery of fjord is incomplete on these dates.

| Date | Equation (3) | | | Equation (2) | | |
|---|---|---|---|---|---|---|
| | Total Volume of Icebergs | Volume of Icebergs: Area > 1000m² | Percent Volume of Icebergs with Area > 1000m² | Total Volume of Icebergs | Volume of Icebergs: Area > 1000m² | Percent Volume of Icebergs with Area > 1000m² |
| Units | km³ | km³ | | km³ | km³ | |
| March 13 | 0.077 | 0.065 | 84 | 0.060 | 0.042 | 70 |
| May 06a | 0.083 | 0.067 | 81 | 0.070 | 0.045 | 65 |
| May 06b* | 0.070 | 0.058 | 83 | 0.056 | 0.036 | 64 |
| June 10a | 0.10 | 0.055 | 53 | 0.12 | 0.042 | 35 |
| June 10b | 0.16 | 0.11 | 67 | 0.16 | 0.071 | 46 |
| July 11* | 0.12 | 0.089 | 72 | 0.11 | 0.054 | 50 |
| July 12a | 0.072 | 0.042 | 58 | 0.077 | 0.029 | 38 |
| July 12b | 0.074 | 0.041 | 56 | 0.079 | 0.029 | 36 |
| November 19 | 0.057 | 0.050 | 88 | 0.042 | 0.031 | 74 |

5    Table 4: Increase in albedo of the entire fjord surface due to the presence of ice and mélange, calculated for each scene. Both icebergs and mélange are taken into account for albedo calculations.  * Imagery of fjord is incomplete on these dates.

| Date | Relative Albedo Increase (%) | Percent Mélange Coverage (%) |
|---|---|---|
| March 13 | 2.5 ± 0.95 | 2.7 ± 0.25 |
| May 06a | 4.8 ± 1.8 | 6.0 ± 0.50 |
| May 06b* | 4.0 ± 1.5 | 2.8 ± 0.24 |
| June 10a | 9.8 ± 3.7 | 11 ± 0.99 |
| June 10b | 9.6 ± 3.6 | 11 ± 0.96 |
| July 11* | 6.2 ± 2.3 | 4.0 ± 0.44 |
| July 12a | 7.7 ± 2.9 | 9.9 ± 0.79 |
| July 12b | 7.4 ± 2.8 | 9.1 ± 0.73 |
| November 19 | 1.2 ± 0.46 | 1.1 ± 0.12 |