# Peer review of "Spatiotemporal Distributions of Icebergs in a Temperate Fjord: Columbia Fjord, Alaska"

_The Cryosphere, 2018_

## Referee Comment (RC1) · Anonymous Referee #1 · 15 Jan 2019

**Review of: Spatiotemporal Distributions of Icebergs in a Temperate Fjord: Columbia Fjord, Alaska**

**General comments:**

This manuscript presents a study of icebergs in the Columbia Fjord, Alaska during March-November 2013. The authors use high resolution satellite images to look at iceberg distribution and size, applying common scaling factors to estimate volume and keel depth. Based on these iceberg metrics, the authors speculate on the influence of icebergs on fjord water properties and note differences and similarities with studies of Greenland fjord icebergs.

The research has succeeded at a basic characterization of Columbia Fjord icebergs. However, the paper falls short on several important fronts.

- The research is motivated (page 2, lines 8-21) by noting an interest in icebergs exiting Columbia Fjord, a topic that is not raised again. A second motivation is that the authors point to Alaska fjords as a proxy for Greenland fjords. However, the discussion of this potential relationship is only vague, failing to provide a reader with a sense of where and when this relationship might hold true.
- The study uses a small sample size, examining only one year. The authors need to provide more information on why only 8 months of data are used. Are images not available from other years? And, if only one year is used, what is the context of this year within the longer periods of observation. Are there reasons to think that this year is dis/similar to other years (e.g., from other published data re: ice discharge, fjord conditions, etc.)? Basing conclusions on such a limited sample provide a weak foundation, so the authors must put in extra work either to increase data or to better contextualize the data that is used.
- Overall, the explanation and discussion does not provide a detailed account of the implications of the study and the meaning of the results. The paper would benefit from more references to existing work, providing context for the study conclusions and discussion. For example, section 4.1 mentions ocean circulation and ocean temperatures, but doesn't provide any of the detail needed for the reader to understand what is known about Columbia Fjord circulation or temperatures. In another example, the discussion in section 4.4 fails to show an appreciation for the wide range of variables and conditions present across Greenland fjords or the many variables involved in the interactions between icebergs and ocean waters. As a result, the discussion is shallow and the conclusions are too general (e.g., see last paragraph in section 4.4).

**Specific comments (by page/line number):**

1/19-21. This is a broad and vague statement regarding Columbia Fjord as a proxy for future Greenland fjords. Given the wide range of variables influencing the role of icebergs in fjords and

the wide variation in physical environments of Greenland, such a general statement is not particularly useful or well substantiated.

2/8. Given how few times Prince Williams Sound is referred to in the paper, it is preferable to use the full phrase rather than an acronym. In general, acronyms make the reader's job more difficult and should be avoided unless for a phrase used widely throughout the paper.

2/19. Here, the authors mention that Columbia Fjord represents a potential analog for future Greenland conditions. Given the wide range of fjord types and conditions in Greenland – and that this is mentioned as a main motivation for the paper – the authors need to be more detailed here. What areas of Greenland might be good candidates? Are there any projections (e.g., of near-Greenland ocean temperatures) that suggest when these analog conditions might occur? As a general statement, it's not very useful.

2/30. What is "Mean Lower Low Water"? This is not something that most glaciologists will be familiar with. This is also another case when the acronym is unnecessary since it's rarely used.

2/35 to 3/1. This sentence does not make sense. Please revise for clarity.

3/15. Instead of "in all but two cases", go ahead and give the information on which cases. The authors are dealing with a relatively small number of images, so it's best to be specific.

3/29. Snow does not float on top of water. Ice mélange is usually considered a conglomerate of icebergs, bergy bits, and growlers, sometimes including sea ice. For the most part, the authors use the phrase correctly, but not in this case. Please correct.

4/9. Introduce the definition of 'melange' when it is first used. It's also odd to discuss bergy bits and growlers in the preceding sentence and then 'tiny chunks of ice' in this sentence.

4/23-28. At no point do the authors explain the use of 'a' and 'b' images. Is there standard area covered by 'a' v. 'b' images? Are they always made into a mosaic? Are areas double counted? A more important point is also raised in this paragraph – the authors attempt to change the threshold for the Nov. 19a image, but are unsuccessful as alleviating the problem. Nevertheless, the authors then continue to include this data in plots and tables. Knowingly poor data should not be included in analysis. The authors have several options – use manual methods to properly characterize the image, remove the data completely, treat this data separately with additional errors, etc.

4/30. In this paragraph, the authors should point to all pertinent figures. For example, point the reader to Fig. 4 for a visual of the 500 m x 500 m squares. Same is true for the first paragraph on page 5.

5/13. Perhaps 'subdividing at 10, 25…". The current sentence is confusing.

5/30. Need to define waterline cross-sectional area at its first use, which is earlier in the paper.

5/30. As best I can tell, the authors do not use waterline cross-sectional area as a proxy for iceberg volume. Instead, they use two methods for determining iceberg volume from waterline cross-sectional area. Later in the paper, they also refer iceberg volume calculated from 'two methods'. Using waterline cross-sectional area would be a third method (and also a worse method than the other two used).

6/7-11. It's not clear why the authors would apply this method for calculating albedo when, I assume, albedo products are available.

6/20-23. The discussion of uncertainty here is an awkward add on to the paragraph. This would be better as a separate section, perhaps combining overall discussion of uncertainty. At minimum, the authors need to discuss the implications of the uncertainties – for example, what percent of the fjord coverage is mélange? Currently, there's no metric for the reader to understand the implications of the numbers given.

6/25. The first half and second half of this sentence say the same thing. Please revise.

7/3. Explain the implications of these numbers.

7/21-25. With such warm water, it seems that iceberg melt would be a substantial component. Is there any published information or other method that the authors can apply to give a sense of the potential magnitude of iceberg melt and its influence?

8/1. The sentence says "evidence *for this*", but the sentence before mentions two possibilities. "This" must be specified.

9/12. What is the area of interest for "all of Columbia Fjord"? Please show on a figure.

Section 4.3. This is a shallow discussion of iceberg influence on freshwater. As some points the authors mention temperature changes, at others they mention salinity. Both are likely affected, but this is not well discussed.

11/2. How can the authors conclude that winter-time capture of ice fragments aren't important when they did not examine winter-time data? The overall discussion in this paragraph also fails to recognize that iceberg residence time is only one factor in how icebergs change ocean water characteristics. For example, ocean water temperature, salinity, and current velocity and direction are also important.

11/19. Why would the icebergs not be significantly rougher than the calving front?

12/4. What evidence is there to attribute the iceberg size change to increased crevasse hydro-fracture? Can the authors cite literature on this or point to observations that suggest this (even qualitatively)?

12/6. How can the authors identify a correlation between anything and average annual calving rate given that they have less than 1 year of data?

12/17. Alaska glaciers contributing to sea level rise is poor justification for studying icebergs in and of itself. I feel the authors need to think more deeply about what is unknown and the most important implications of their work.

Figure 1. It would be useful to label Heather Moraine in the image. 'PWS' label in c. is unnecessary. Is it possible to add the outline of the region in d. to the c. panel?

Figures 4 and 8. It would be nice to put this data on a map/image background.

Figure 5. Why not show data from all periods? Also, it would be preferable to list the class sizes in the captions.

Figure 6. Bad data should not be included in the plot.

Table 2. What is the source of these errors? What does this error represent?

Tables 3 and 4. Again, bad data should not be included in plots or tables. Either fix it or remove it.

**Technical corrections (by page/line number):**

2/20. Please remove "in order" for all cases of "in order to". It is unnecessary.

---

## Referee Comment (RC2) · Anonymous Referee #2 · 22 Jan 2019

Review of tc-2018-230 Spatiotemporal Distributions of Icebergs in a Temperate Fjord: Columbia Fjord, Alaska by S. Neuhaus et al

General comment

The paper analyzes the distribution of icebergs within the Columbia fjord using high resolution (0.5 m) visible satellite images (10 covering a 8 months period in 2013). Classical image processing tools (from Matlab) are then used to detect and estimate the size of the icebergs within the fjord. The results are validated by comparison with manual analysis for selected scenes. The results of the images analysis are then used to compute the time/space distribution of iceberg and the evolution of the distribution

of the icebergs' size in time and location within the fjord. The analysis reveals the complex distributions of the iceberg within the fjord as well as the seasonal variability related to the glacier calving rate.

The results are of interest and are worth publishing. However, the motivation of the study should be stated more clearly and if the implication for the future of Greenland icebergs (if kept) should be better justified.

The study is also limited to 8 month and 10 images where certainly much more are available. I know that image processing is quite hard and fastidious but at least the authors should justified why they limited their study to this short data set. Paragraph 4.3 and 4.4 need to be better focused on real results and not on quite shallow general discussions. The distribution of volume and the evolution of the size distribution are important results by themselves. I think that the study could also be improved if simple computation of freshwater flux using ice volume and classical melting law were conducted and presented.

Specific comments

Page 1 line 8 outet=> outlet

Page 1 line 20: Coloumbia==> Columbia

Page 1 line 20 Considering the difference of temperature between Greenland and PWS water and the different conditions of the Greenland fjords this remark is certainly way to general.

Page 3 lines 15-20. For people not familiar with WorldView Satellite explain why there are sometimes two images from the same satellite at the same time. The sentence on the hundredths of second separation is quite useless.

Page 3 line 23 and following; As the study fully relies on the detection and analysis of the WW1 images, it is important to at least present an example of detection (on an image detail) at best to provide the analysis of all images in Supplementary Information.

Page 4 lines 23-26. It is not explain what is the difference between a and b images ( see my previous comment). If this is related to different viewing angle it is important to precise it as it might explain the different result (that could be due to a difference in effective resolution). I don't understand the November 19 case (not enough information). There again it could be related to viewing angle and specular reflection on open water (wild guess as we don't have the data and there are not freely accessible).

Page 4 line 32. Why May 06 (a b) is not included? Provide explanation. reference to figure 4 should be included.

Page 5 line 13-14, The sentence is not very clear. The pdf is computed on the following bins.

Page 5 &2.4. This paragraph presents two methods of estimation of the iceberg volume from the satellite iceberg area (which might be different from the waterline cross-sectional area if the water is very clear). A is not a proxy.

Page 6 line 7-. I think the authors recompute the albedo using fixed ocean and ice value to eliminate solar angle and atmospheric influence but it is not stated.

Page 6 lin18-20 Where is the 95% coming from. Figure 4 and 5 don't present proportion but numbers and area. Figure 4 should use a log colorscale to reveal more details of the distribution within the fjord.

Page 6 line 20-24. I don't understand the purpose of this remark. It is part of the detection and analysis method and should be treated there.

Page 6 line 25. Please mention figure 6 from the beginning of the &.

Figure 6. For May 6 b and Jul 11 there are only estimates for the proximal zone certainly because of the partial coverage. Is it really necessary to keep those two images as they don't really bring any special information/results.

Page 7 line 13-15. The comparison of the thickness for a given A using 2 and 3 explains
completely the difference observed in Table 3. What is important to note here is that the variations of the total volume, and percentage of volume for large icebergs are very similar using the two formulas although they give very different iceberg thickness (this is certainly due to the strong impact of the power-law distribution of the distribution of the ice volume).

Page 7 line 19. The summer increase of albedo could implies that the fragmentation is increasing in warmer waters.

Page 7 line 21-25. It is important to explain in detail the computation of the residence time.

Page 8 line 23-25 In fact power-law and lognormal are quite similar and power-laws (which do not converge (tend to infinity in 0) ) can be used to approximated the tail of lognormal distributions.

Page 9 line 7-17 Two recent studies one from Bouhier et al (https://doi.org/10.5194/tc-12-2267-2018) and one from Crawford et al (https://doi.org/10.1029/2018JC014388) presented size distributions of pieces resulting from icebergs fragmentation with slope close to -1.5 (i.e. the mid_fjord summer slope). The two studies mentioned that this -1.5 slope is in general associated to fragile fragmentation and could in this case indicate that during summer month the main driver of the size distribution within the fjord is the fragmentation.

Page 9 line 26-27 In fact when computing power-law distribution there is always some problem with the tail of the distribution just because the numbers of samples is too low.

Page 9 line 30. This is a direct consequence of the power law distribution.

Page 9-10 &4.3 This paragraph is not very clear and don't present any significant results. It could be of interest if the volume of ice and melting law were used to estimate the fresh water flux from each image.

Page 10 &4.4 There is no data in winter in your study.

Page 11 line 1 Where is 11% mélange coming from (not from Table4).

Page 11 line 14-15. The computation of the iceberg surface is not obvious. It should be given (in annex).

Page 12 line 4-6. The -1.5 slope could indicate that there is more fragmentation during summer.

---

## Author Comment (AC1) · 6 Mar 2019

**Reviewer #1**

**Review of: Spatiotemporal Distributions of Icebergs in a Temperate Fjord: Columbia Fjord, Alaska**

**General comments:**

This manuscript presents a study of icebergs in the Columbia Fjord, Alaska during March- November 2013. The authors use high resolution satellite images to look at iceberg distribution and size, applying common scaling factors to estimate volume and keel depth. Based on these iceberg metrics, the authors speculate on the influence of icebergs on fjord water properties and note differences and similarities with studies of Greenland fjord icebergs.

The research has succeeded at a basic characterization of Columbia Fjord icebergs. However, the paper falls short on several important fronts.

- - The research is motivated (page 2, lines 8-21) by noting an interest in icebergs exiting Columbia Fjord, a topic that is not raised again. A second motivation is that the authors point to Alaska fjords as a proxy for Greenland fjords. However, the discussion of this potential relationship is only vague, failing to provide a reader with a sense of where and when this relationship might hold true.

- - The study uses a small sample size, examining only one year. The authors need to provide more information on why only 8 months of data are used. Are images not available from other years? And, if only one year is used, what is the context of this year within the longer periods of observation. Are there reasons to think that this year is dis/similar to other years (e.g., from other published data re: ice discharge, fjord conditions, etc.)? Basing conclusions on such a limited sample provide a weak foundation, so the authors must put in extra work either to increase data or to better contextualize the data that is used.

- - Overall, the explanation and discussion does not provide a detailed account of the implications of the study and the meaning of the results. The paper would benefit from more references to existing work, providing context for the study conclusions and discussion. For example, section 4.1 mentions ocean circulation and ocean temperatures, but doesn't provide any of the detail needed for the reader to understand what is known about Columbia Fjord circulation or temperatures. In another example, the discussion in section 4.4 fails to show an appreciation for the wide range of variables and conditions present across Greenland fjords or the many variables involved in the interactions between icebergs and ocean waters. As a result, the discussion is shallow and the conclusions are too general (e.g., see last paragraph in section 4.4).

**Specific comments (by page/line number):**

1/19-21. This is a broad and vague statement regarding Columbia Fjord as a proxy for future Greenland fjords. Given the wide range of variables influencing the role of

icebergs in fjords and the wide variation in physical environments of Greenland, such a general statement is not particularly useful or well substantiated.

*Authors' Response: Thank you for pointing the fact that our comparison of Columbia Fjord to Greenland fjords is not as compelling as it could be. We are working on correcting this in the revised version of the manuscript.*

2/8. Given how few times Prince Williams Sound is referred to in the paper, it is preferable to use the full phrase rather than an acronym. In general, acronyms make the reader's job more difficult and should be avoided unless for a phrase used widely throughout the paper.

*Authors' Response: We agree that acronyms can be confusing. We have changed all mentions of "PWS" to Prince William Sound to avoid confusion.*

2/19. Here, the authors mention that Columbia Fjord represents a potential analog for future Greenland conditions. Given the wide range of fjord types and conditions in Greenland – and that this is mentioned as a main motivation for the paper – the authors need to be more detailed here. What areas of Greenland might be good candidates? Are there any projections (e.g., of near-Greenland ocean temperatures) that suggest when these analog conditions might occur? As a general statement, it's not very useful.

*Authors' Response: As stated above, we agree that our comparison to Greenland fjords is too general. We appreciate the reviewer highlighting the weakness of this section, and we are working on addressing this in our revisions.*

2/30. What is "Mean Lower Low Water"? This is not something that most glaciologists will be familiar with. This is also another case when the acronym is unnecessary since it's rarely used.

*Authors' Response: Mean Lower Low Water is the elevation datum that the National Oceanographic and Atmospheric Administration (NOAA) uses to reference tidal height measurements. We keep the mention of Mean Lower Low Water because it is the defined standard elevation of zero above/below sea level, however we have added in a statement clarifying the definition to those who may not be familiar with it.*

2/35 to 3/1. This sentence does not make sense. Please revise for clarity.

*Authors' Response: We have split this convoluted sentence into two simpler sentences for clarity.*

3/15. Instead of "in all but two cases", go ahead and give the information on which cases. The authors are dealing with a relatively small number of images, so it's best to be specific.

*Authors' Response: We have removed "in all but two cases" and instead added in the image names so as to avoid confusion.*

3/29. Snow does not float on top of water. Ice mélange is usually considered a conglomerate of icebergs, bergy bits, and growlers, sometimes including sea ice. For the most part, the authors use the phrase correctly, but not in this case. Please correct.

*Authors' Response: We have removed the mention of snow from this sentence, as recommended. Instead, we have amended our definition of ice mélange and moved it to this place in the manuscript, because it is the first time we mention ice mélange.*

4/9. Introduce the definition of 'melange' when it is first used. It's also odd to discuss bergy bits and growlers in the preceding sentence and then 'tiny chunks of ice' in this sentence.

*Authors' Response: We have moved this definition of ice mélange to the location in the paper where we first mention mélange.*

4/23-28. At no point do the authors explain the use of 'a' and 'b' images. Is there standard area covered by 'a' v. 'b' images? Are they always made into a mosaic? Are areas double counted? A more important point is also raised in this paragraph – the authors attempt to change the threshold for the Nov. 19a image, but are unsuccessful as alleviating the problem. Nevertheless, the authors then continue to include this data in plots and tables. Knowingly poor data should not be included in analysis. The authors have several options – use manual methods to properly characterize the image, remove the data completely, treat this data separately with additional errors, etc.

*Authors' Response: For a few of the dates for which we have repeat images, there are two images labelled 'a' and 'b'. These images are covering the same area from somewhat different angles and were taken by the satellite for the purpose of stereo imagery. They were taken approximately one minute apart. We are able to use both images, and thus label them 'a' and 'b'. We did not explicitly state this in the manuscript previously, so we have now corrected this by adding in a few clarifying sentences.*

*Regarding the November 19a imagery, we previously did know about the quality problems with identifying icebergs in this image but decided that there is value in including these flawed data. However, after reading the reviewer's comment, we decided to discard the November 19a dataset completely because we do agree that the problems with data quality do not justify including this dataset in our analyses.*

4/30. In this paragraph, the authors should point to all pertinent figures. For example, point the reader to Fig. 4 for a visual of the 500 m x 500 m squares. Same is true for the first paragraph on page 5.

*Authors' Response: Thank you for this suggestion, we have fixed this by adding in references at the appropriate locations.*

5/13. Perhaps 'subdividing at 10, 25...". The current sentence is confusing.

*Authors' Response: We have re-written this sentence to make it less confusing.*

5/30. Need to define waterline cross-sectional area at its first use, which is earlier in the paper.

*Authors' Response: We have moved our definition of waterline cross-sectional area to the place in the manuscript where we first make use of this term.*

5/30. As best I can tell, the authors do not use waterline cross-sectional area as a proxy for iceberg volume. Instead, they use two methods for determining iceberg volume from waterline cross-sectional area. Later in the paper, they also refer iceberg volume calculated from 'two methods'. Using waterline cross-sectional area would be a third method (and also a worse method than the other two used).

*Authors' Response: We do not use waterline cross-sectional area as a proxy for iceberg volume as we actually calculate iceberg volume using two different methods. We have removed the mention of using waterline cross-sectional area as a proxy.*

6/7-11. It's not clear why the authors would apply this method for calculating albedo when, I assume, albedo products are available.

*Authors' Response: We calculated albedo using this method because the ocean albedo is highly dependent on the angle of incoming solar radiation. In addition, the atmosphere can affect surface albedo measurements made from satellite imagery. We therefore chose this method to avoid these issues and to focus the discussion in our manuscript on the direct contribution of icebergs to the surface albedo. We have added in a sentence stating our reasoning for calculating albedo in this manner.*

6/20-23. The discussion of uncertainty here is an awkward add on to the paragraph. This would be better as a separate section, perhaps combining overall discussion of uncertainty. At minimum, the authors need to discuss the implications of the uncertainties – for example, what percent of the fjord coverage is mélange? Currently, there's no metric for the reader to understand the implications of the numbers given.

*Authors' Response: We appreciate the reviewer bringing this to our attention. We are working on fleshing out our discussion of uncertainties associated with iceberg identification in areas of mélange.*

6/25. The first half and second half of this sentence say the same thing. Please revise.

*Authors' Response: We have revised the sentence to make it less repetitive.*

7/3. Explain the implications of these numbers.

*Authors' Response: We are slightly confused by this comment. Since this is the Results section, we focus on specifying by how much the power exponent of iceberg size distribution varies between the three major sections of the fjord. We discuss the implications of the numbers presented here in the discussion section.*

7/21-25. With such warm water, it seems that iceberg melt would be a substantial component. Is there any published information or other method that the authors can apply to give a sense of the potential magnitude of iceberg melt and its influence?

*Authors' Response: We agree that given the temperate conditions of this fjord, it would be a good idea to discuss iceberg melt. We are working to elucidate the impact of melting on iceberg distribution and evolution in the fjord.*

8/1. The sentence says "evidence *for this*", but the sentence before mentions two possibilities. "This" must be specified.

*Authors' Response: We have added in a statement clarifying what is meant here. Thank you for catching this problem.*

9/12. What is the area of interest for "all of Columbia Fjord"? Please show on a figure.

*Authors' Response: What is meant by "all of Columbia Fjord" is the entire fjord as defined in the introduction. We are working to make this more clear in the manuscript text.*

Section 4.3. This is a shallow discussion of iceberg influence on freshwater. As some points the authors mention temperature changes, at others they mention salinity. Both are likely affected, but this is not well discussed.

*Authors' Response: We are working to address these shortcomings.*

11/2. How can the authors conclude that winter-time capture of ice fragments aren't important when they did not examine winter-time data? The overall discussion in this paragraph also fails to recognize that iceberg residence time is only one factor in how icebergs change ocean water characteristics. For example, ocean water temperature, salinity, and current velocity and direction are also important.

*Authors' Response: We have imagery from March 13th, which is before the vernal equinox, i.e., the formal end of the astronomical winter. We do realize that it is standard in meteorological and climatic analyses to consider only December, January, and February to be winter months. However, the winter maximum of Arctic sea ice extent is typically reached in March, often around the time of when our March 13th scene was acquired. Hence, we do not think that it is too much of a stretch to consider this scene to be representative of winter time iceberg conditions in Columbia Fjord. We agree with the reviewer that this section does not address some important ways in which icebergs may affect fjord waters. We are working to address this.*

11/19. Why would the icebergs not be significantly rougher than the calving front?

*Authors' Response: Because both icebergs and the terminus are made of the same material and exposed to the same ocean water, we assumed that they would exhibit similar roughness. In our calculation of iceberg and glacier terminus surface area we made the simplifying assumption that they have the same roughness, given that we have no observational constraints on roughness for either surface. We will discuss this assumption in more detail in the context of current literature on this topic.*

12/4. What evidence is there to attribute the iceberg size change to increased crevasse hydro- fracture? Can the authors cite literature on this or point to observations that suggest this (even qualitatively)?

*Authors' Response: Thank you for highlighting the problem with our discussions of this issue. Several previous studies have shown that power-law exponents of -1.5 are an indication of brittle fragmentation. The summer power-law exponent values we find in this study are closer to -1.5 in the summer than in the spring or fall. This seasonal evolution of the power-law exponent supports the proposition that hydro fracture is an important process in summer, when surface melt rates are high and water-filled fractures should be abundant. We have improved our discussion of these issues in the revised manuscript in response to this comment of the reviewer.*

12/6. How can the authors identify a correlation between anything and average annual calving rate given that they have less than 1 year of data?

*Authors' Response: Columbia Glacier velocity, flux, and terminus position data presented in reports from the US Geological Survey and Vijay and Braun (2017) indicate that 2013 was a reasonably representative year in terms of iceberg calving rates. We therefore feel it is justified to compare our observations of icebergs in Columbia Fjord in 2013 to an average annual calving rate that was calculated for 2013.*

12/17. Alaska glaciers contributing to sea level rise is poor justification for studying icebergs in and of itself. I feel the authors need to think more deeply about what is unknown and the most important implications of their work.

*Authors' Response: We are working to address this comment.*

Figure 1. It would be useful to label Heather Moraine in the image. 'PWS' label in c. is unnecessary. Is it possible to add the outline of the region in d. to the c. panel?

*Authors' Response: We had previously outlined Heather Moraine in panel c, however, we have now added a label to make our outline more clear and noticeable to the reader. We have removed PWS from panel c. Instead of outlining the region covered by panel d in panel c, we have moved that bathymetry shown in panel d so that it overlaps the fjord extent shown in panel c.*

Figures 4 and 8. It would be nice to put this data on a map/image background.

*Authors' Response: We have added in a satellite image of the fjord taken on June 10, 2013 to the background of figures 4 and 8 to give the reader further spatial context.*

Figure 5. Why not show data from all periods? Also, it would be preferable to list the class sizes in the captions.

*Authors' Response: We agree that it would be preferable to show the data from all dates. We have included plots from the other images.*

Figure 6. Bad data should not be included in the plot.

*Authors' Response: We assume the reviewer is referring to the November 19a data. We have removed this data from the plot entirely.*

Table 2. What is the source of these errors? What does this error represent?

*Authors' Response: The errors are the upper and lower limits of the 95% confidence interval.*

Tables 3 and 4. Again, bad data should not be included in plots or tables. Either fix it or remove it.

*Authors' Response: We assume the reviewer is referring to the data from November 19a. We have removed this data from both tables.*

**Technical corrections (by page/line number):**

2/20. Please remove "in order" for all cases of "in order to". It is unnecessary.

*Authors' Response: We have changed all cases of "in order to" to simply "to."*

---

## Author Comment (AC2) · 6 Mar 2019

Anonymous Referee #2 Review of tc-2018-230 Spatiotemporal Distributions of Icebergs in a Temperate Fjord: Columbia Fjord, Alaska by S. Neuhaus et al General comment The paper analyzes the distribution of icebergs within the Columbia fjord using high resolution (0.5 m) visible satellite images (10 covering a 8 months period in 2013). Classical image processing tools (from Matlab) are then used to detect and estimate the size of the icebergs within the fjord. The results are validated by comparison with manual analysis for selected scenes. The results of the images analysis are then used to compute the time/space distribution of iceberg and the evolution of

the distribution of the icebergs' size in time and location within the fjord. The analysis reveals the complex distributions of the iceberg within the fjord as well as the seasonal variability related to the glacier calving rate. The results are of interest and are worth publishing. However, the motivation of the study should be stated more clearly and if the implication for the future of Greenland icebergs (if kept) should be better justified. The study is also limited to 8 month and 10 images where certainly much more are available. I know that image processing is quite hard and fastidious but at least the authors should justified why they limited their study to this short data set. Paragraph 4.3 and 4.4 need to be better focused on real results and not on quite shallow general discussions. The distribution of volume and the evolution of the size distribution are important results by themselves. I think that the study could also be improved if simple computation of freshwater flux using ice volume and classical melting law were conducted and presented.

Specific comments

Page 1 line 8 outet=> outlet

Authors' Response: We have corrected this typo.

- Page 1 line 20: Coloumbia==> Columbia

Authors' Response: We appreciate that the reviewer caught this typo as well. We have corrected it.

- Page 1 line 20 Considering the difference of temperature between Greenland and PWS water and the different conditions of the Greenland fjords this remark is certainly way to general.

Authors' Response: We recognize that our comparison of Prince William Sound in Alaska and fjords in Greenland was far too general. We are currently working on strengthening it and making it more specific.

- Page 3 lines 15-20. For people not familiar with WorldView Satellite explain why there

are sometimes two images from the same satellite at the same time. The sentence on the hundredths of second separation is quite useless.

Authors' Response: We agree that there should be an explanation of why the World-View Satellite would take two images only minutes apart. The images were taken for the purposes of stereo imaging of the same area from slightly different angles. We have thus added in a few sentences in Section 2.1 to explain this. We have also removed the sentence mentioning "hundredths of a second" because it is unhelpful.

- Page 3 line 23 and following; As the study fully relies on the detection and analysis of the WW1 images, it is important to at least present an example of detection (on an image detail) at best to provide the analysis of all images in Supplementary Information.

Authors' Response: To give readers a clearer idea of what we have done, we have added in an example of iceberg detection to figure 2, as well as adding in an explanatory sentence referencing said figure.

- Page 4 lines 23-26. It is not explain what is the difference between a and b images ( see my previous comment). If this is related to different viewing angle it is important to precise it as it might explain the different result (that could be due to a difference in effective resolution). I don't understand the November 19 case (not enough information). There again it could be related to viewing angle and specular reflection on open water (wild guess as we don't have the data and there are not freely accessible).

Authors' Response: We have now added in a few sentences distinguishing 'a' and 'b' images to the text, as well as mentioning that the difference in iceberg identification could be due to viewing angle. Changing the angle at which we view the ocean changes the reflectivity of the ocean, which could therefore affect which pixels were identified as ice versus water.

- Page 4 line 32. Why May 06 (a b) is not included? Provide explanation. reference to figure 4 should be included.

Authors' Response: Omitting May 06a from the figure was a mistake on our part. We have fixed this by adding the icebergs from May 06a into figures 4a and 4b. We have also added references in the text to these two figures.

- Page 5 line 13-14, The sentence is not very clear. The pdf is computed on the following bins.

Authors' Response: We have re-worded this sentence to make it less confusing.

- Page 5 &2.4. This paragraph presents two methods of estimation of the iceberg volume from the satellite iceberg area (which might be different from the waterline cross- sectional area if the water is very clear). A is not a proxy.

Authors' Response: The reviewer is correct about our mis-statement on this topic. We do not use waterline cross-sectional area as a proxy for iceberg volume as we actually calculate iceberg volume using two different methods. We have removed the mention of using waterline cross-sectional area as a proxy.

- Page 6 line 7-. I think the authors recompute the albedo using fixed ocean and ice value to eliminate solar angle and atmospheric influence but it is not stated.

Authors' Response: Yes, we calculated the albedo using this method in order to avoid issues with atmospheric interference as well as the angle at which solar radiation hits the ocean. Our objective is to focus on the direct impact of iceberg coverage on albedo. We have added in a sentence stating this more clearly.

- Page 6 lin18-20 Where is the 95% coming from. Figure 4 and 5 don't present proportion but numbers and area. Figure 4 should use a log colorscale to reveal more details of the distribution within the fjord.

Authors' Response: 95% of all the icebergs identified in this study had a waterline cross-sectional area less than or equal to 100 m2. As this is not readily discernible from the figure, we are amending it, as well as re-wording the text to make this clearer.

- Page 6 line 20-24. I don't understand the purpose of this remark. It is part of the detection and analysis method and should be treated there.

Authors' Response: We have moved this to the methods section.

- Page 6 line 25. Please mention figure 6 from the beginning of the &.

Authors' Response: We have added in a reference to figure 6 in the first sentence of the paragraph.

- Figure 6. For May 6 b and Jul 11 there are only estimates for the proximal zone certainly because of the partial coverage. Is it really necessary to keep those two images as they don't really bring any special information/results.

Authors' Response: Despite the fact that there is incomplete coverage for the fjord on May 06b and July 11, we believe there is value in representing the partial data in the figure. The power-law exponents for these two dates show agreement with the power-law exponents in the proximal zone for the other dates in the same seasons.

- Page 7 line 13-15. The comparison of the thickness for a given A using 2 and 3 explains completely the difference observed in Table 3. What is important to note here is that the variations of the total volume, and percentage of volume for large icebergs are very similar using the two formulas although they give very different iceberg thickness (this is certainly due to the strong impact of the power-law distribution of the distribution of the ice volume).

Authors' Response: This is a very good point. We are working on addressing this in the text.

- Page 7 line 19. The summer increase of albedo could implies that the fragmentation is increasing in warmer waters.

Authors' Response: This is an interesting point. We have added a sentence addressing this into the discussion section.

- Page 7 line 21-25. It is important to explain in detail the computation of the residence time.

Authors' Response: We have clarified our computation of iceberg residence time by stating our methods more explicitly.

- Page 8 line 23-25 In fact power-law and lognormal are quite similar and power-laws (which do not converge (tend to infinity in 0) ) can be used to approximated the tail of lognormal distributions.

Authors' Response: This is an interesting point, and something we are working to address in the manuscript.

- Page 9 line 7-17 Two recent studies one from Bouhier et al (https://doi.org/10.5194/tc-12-2267-2018) and one from Crawford et al (https://doi.org/10.1029/2018JC014388) presented size distributions of pieces resulting from icebergs fragmentation with slope close to -1.5 (i.e. the mid_fjord summer slope). The two studies mentioned that this -1.5 slope is in general associated to fragile fragmentation and could in this case indicate that during summer month the main driver of the size distribution within the fjord is the fragmentation.

Authors' Response: We very much appreciate the reviewer bringing these two studies to our attention. We have used their findings, as well as those from other studies of iceberg fragmentation, to expand our discussion of brittle fragmentation. We agree that the power-law exponents we find in this study indicate that there is more brittle fragmentation of icebergs during the summer months than the spring or fall. We have added in a number of sentences to the discussion on this topic.

- Page 9 line 26-27 In fact when computing power-law distribution there is always some problem with the tail of the distribution just because the numbers of samples is too low. Authors' Response: The reviewer makes a very good point. We have added in a sentence explaining this in the methods section where we first mention removing the

tail of the distribution to achieve a better fit.

Page 9 line 30. This is a direct consequence of the power law distribution.

Authors' Response: We agree with this point.

- Page 9-10 &4.3 This paragraph is not very clear and don't present any significant results. It could be of interest if the volume of ice and melting law were used to estimate the fresh water flux from each image.

Authors' Response: We recognize that this paragraph is too general and does not present significant results. We are currently addressing this to include estimations of freshwater input.

- Page 10 &4.4 There is no data in winter in your study.

Authors' Response: We have imagery from March 13th, which is before the vernal equinox, i.e., the formal end of the astronomical winter. We do realize that it is standard in meteorological and climatic analyses to consider only December, January, and February to be winter months. However, the winter maximum of Arctic sea ice extent is typically reached in March, often around the time of when our March 13th scene was acquired. Hence, we do not think that it is too much of a stretch to consider this scene to be representative of winter time iceberg conditions in Columbia Fjord. We agree with the reviewer that this section does not address some important ways in which icebergs may affect fjord waters. We are working to address this.

- Page 11 line 1 Where is 11% mélange coming from (not from Table4).

Authors' Response: We calculated the mélange by subtracting the total number of ice pixels by the number of ice pixels that are identified as part of an iceberg. We recognize that we do not state this explicitly and are working to correct this.

- Page 11 line 14-15. The computation of the iceberg surface is not obvious. It should be given (in annex).

Authors' Response: We agree that we do not explicitly explain our calculations of iceberg surface area. We are correcting this.

- Page 12 line 4-6. The -1.5 slope could indicate that there is more fragmentation during summer.

Authors' Response: We agree with the reviewer on this point. We have therefore edited this sentence to include brittle fragmentation.

---

## Author Comment (AC3) · 14 Mar 2019

**Reviewer #1**

**Review of: Spatiotemporal Distributions of Icebergs in a Temperate Fjord: Columbia Fjord, Alaska**

**General comments:**

This manuscript presents a study of icebergs in the Columbia Fjord, Alaska during March-November 2013. The authors use high resolution satellite images to look at iceberg distribution and size, applying common scaling factors to estimate volume and keel depth. Based on these iceberg metrics, the authors speculate on the influence of icebergs on fjord water properties and note differences and similarities with studies of Greenland fjord icebergs.

The research has succeeded at a basic characterization of Columbia Fjord icebergs. However, the paper falls short on several important fronts.

- - The research is motivated (page 2, lines 8-21) by noting an interest in icebergs exiting Columbia Fjord, a topic that is not raised again. A second motivation is that the authors point to Alaska fjords as a proxy for Greenland fjords. However, the discussion of this potential relationship is only vague, failing to provide a reader with a sense of where and when this relationship might hold true.

- - The study uses a small sample size, examining only one year. The authors need to provide more information on why only 8 months of data are used. Are images not available from other years? And, if only one year is used, what is the context of this year within the longer periods of observation. Are there reasons to think that this year is dis/similar to other years (e.g., from other published data re: ice discharge, fjord conditions, etc.)? Basing conclusions on such a limited sample provide a weak foundation, so the authors must put in extra work either to increase data or to better contextualize the data that is used.

- - Overall, the explanation and discussion does not provide a detailed account of the implications of the study and the meaning of the results. The paper would benefit from more references to existing work, providing context for the study conclusions and discussion. For example, section 4.1 mentions ocean circulation and ocean temperatures, but doesn't provide any of the detail needed for the reader to understand what is known about Columbia Fjord circulation or temperatures. In another example, the discussion in section 4.4 fails to show an appreciation for the wide range of variables and conditions present across Greenland fjords or the many variables involved in the interactions between icebergs and ocean waters. As a result, the discussion is shallow and the conclusions are too general (e.g., see last paragraph in section 4.4).

*Authors' Response: We would like to thank the reviewer for drawing to our attention the fact that our discussion of Columbia Fjord as a potential future Greenlandic fjord is too general. We have made sure to address this in our revisions. Instead of justifying our study by presenting Columbia Fjord as a potential future Greenland*

*fjord, we have focused our manuscript on the more specific implications of our results. We have amended section 4.4 so that it is now a more detailed comparison of our findings to measurements which have been previously published in papers covering the topic of icebergs in fjords.*

*We were able to obtain satellite imagery of Columbia Fjord for one year (2013). To check if this year was reasonably representative of a longer period of time, we examined publications by the US Geological Survey and Vijay and Braun (2017), both of which presented data on iceberg calving rates for multiple years and indicated that 2013 was a fairly typical year for iceberg production in Columbia Fjord.*

**Specific comments (by page/line number):**

1/19-21. This is a broad and vague statement regarding Columbia Fjord as a proxy for future Greenland fjords. Given the wide range of variables influencing the role of icebergs in fjords and the wide variation in physical environments of Greenland, such a general statement is not particularly useful or well substantiated.

*Authors' Response: Thank you for pointing the fact that our comparison of Columbia Fjord to Greenland fjords is not as compelling as it could be. We have removed our statements of Columbia Fjord as a proxy for future Greenlandic fjords.*

2/8. Given how few times Prince Williams Sound is referred to in the paper, it is preferable to use the full phrase rather than an acronym. In general, acronyms make the reader's job more difficult and should be avoided unless for a phrase used widely throughout the paper.

*Authors' Response: We agree that acronyms can be confusing. We have changed all mentions of "PWS" to Prince William Sound to avoid confusion.*

2/19. Here, the authors mention that Columbia Fjord represents a potential analog for future Greenland conditions. Given the wide range of fjord types and conditions in Greenland – and that this is mentioned as a main motivation for the paper – the authors need to be more detailed here. What areas of Greenland might be good candidates? Are there any projections (e.g., of near-Greenland ocean temperatures) that suggest when these analog conditions might occur? As a general statement, it's not very useful.

*Authors' Response: As stated above, we agree that our comparison to Greenland fjords is too general. We have amended this by instead focusing on comparing our results to other results from previous studies of icebergs in Greenlandic fjords.*

2/30. What is "Mean Lower Low Water"? This is not something that most glaciologists will be familiar with. This is also another case when the acronym is unnecessary since it's rarely used.

*Authors' Response: Mean Lower Low Water is the elevation datum that the National Oceanographic and Atmospheric Administration (NOAA) uses to reference tidal height measurements. We keep the mention of Mean Lower Low Water because it is the defined standard elevation of zero above/below sea level, however we have added in a statement clarifying the definition to those who may not be familiar with it.*

2/35 to 3/1. This sentence does not make sense. Please revise for clarity.

*Authors' Response: We have split this convoluted sentence into two simpler sentences for clarity.*

3/15. Instead of "in all but two cases", go ahead and give the information on which cases. The authors are dealing with a relatively small number of images, so it's best to be specific.

*Authors' Response: We have removed "in all but two cases" and instead added in the image names so as to avoid confusion.*

3/29. Snow does not float on top of water. Ice mélange is usually considered a conglomerate of icebergs, bergy bits, and growlers, sometimes including sea ice. For the most part, the authors use the phrase correctly, but not in this case. Please correct.

*Authors' Response: We have removed the mention of snow from this sentence, as recommended. Instead, we have amended our definition of ice mélange and moved it to this place in the manuscript, because it is the first time we mention ice mélange.*

4/9. Introduce the definition of 'melange' when it is first used. It's also odd to discuss bergy bits and growlers in the preceding sentence and then 'tiny chunks of ice' in this sentence.

*Authors' Response: We have moved this definition of ice mélange to the location in the paper where we first mention mélange.*

4/23-28. At no point do the authors explain the use of 'a' and 'b' images. Is there standard area covered by 'a' v. 'b' images? Are they always made into a mosaic? Are areas double counted? A more important point is also raised in this paragraph – the authors attempt to change the threshold for the Nov. 19a image, but are unsuccessful as alleviating the problem. Nevertheless, the authors then continue to include this data in plots and tables. Knowingly poor data should not be included in analysis. The authors have several options – use manual methods to properly characterize the image, remove the data completely, treat this data separately with additional errors, etc.

*Authors' Response: For a few of the dates for which we have repeat images, there are two images labelled 'a' and 'b'. These images are covering the same area from somewhat different angles and were taken by the satellite for the purpose of stereo imagery. They were taken approximately one minute apart. We are able to use both images, and thus label them 'a' and 'b'. We did not explicitly state this in the manuscript previously, so we have now corrected this by adding in a few clarifying sentences.*

*Regarding the November 19a imagery, we previously did know about the quality problems with identifying icebergs in this image but decided that there is value in including these flawed data. However, after reading the reviewer's comment, we decided to discard the November 19a dataset completely because we do agree that the problems with data quality do not justify including this dataset in our analyses.*

4/30. In this paragraph, the authors should point to all pertinent figures. For example, point the reader to Fig. 4 for a visual of the 500 m x 500 m squares. Same is true for the first paragraph on page 5.

*Authors' Response: Thank you for this suggestion, we have fixed this by adding in references at the appropriate locations.*

5/13. Perhaps 'subdividing at 10, 25...". The current sentence is confusing.

*Authors' Response: We have re-written this sentence to make it less confusing.*

5/30. Need to define waterline cross-sectional area at its first use, which is earlier in the paper.

*Authors' Response: We have moved our definition of waterline cross-sectional area to the place in the manuscript where we first make use of this term.*

5/30. As best I can tell, the authors do not use waterline cross-sectional area as a proxy for iceberg volume. Instead, they use two methods for determining iceberg volume from waterline cross-sectional area. Later in the paper, they also refer iceberg volume calculated from 'two methods'. Using waterline cross-sectional area would be a third method (and also a worse method than the other two used).

*Authors' Response: We do not use waterline cross-sectional area as a proxy for iceberg volume as we actually calculate iceberg volume using two different methods. We have removed the mention of using waterline cross-sectional area as a proxy.*

6/7-11. It's not clear why the authors would apply this method for calculating albedo when, I assume, albedo products are available.

*Authors' Response: We calculated albedo using this method because the ocean albedo is highly dependent on the angle of incoming solar radiation. In addition, the atmosphere can affect surface albedo measurements made from satellite imagery. We therefore chose this method to avoid these issues and to focus the discussion in our manuscript on the direct contribution of icebergs to the surface albedo. We have added in a sentence stating our reasoning for calculating albedo in this manner.*

6/20-23. The discussion of uncertainty here is an awkward add on to the paragraph. This would be better as a separate section, perhaps combining overall discussion of uncertainty. At minimum, the authors need to discuss the implications of the uncertainties – for example, what percent of the fjord coverage is mélange? Currently, there's no metric for the reader to understand the implications of the numbers given.

*Authors' Response: We appreciate the reviewer bringing this to our attention. We have moved the discussion of uncertainty associated with iceberg identification in areas of mélange to the methods section. In addition, we have presented mélange coverage as a percentage of fjord area for the dates we examined.*

6/25. The first half and second half of this sentence say the same thing. Please revise.

*Authors' Response: We have revised the sentence to make it less repetitive.*

7/3. Explain the implications of these numbers.

*Authors' Response: We are slightly confused by this comment. Since this is the Results section, we focus on specifying by how much the power exponent of iceberg size distribution varies between the three major sections of the fjord. We discuss the implications of the numbers presented here in the discussion section.*

7/21-25. With such warm water, it seems that iceberg melt would be a substantial component. Is there any published information or other method that the authors can apply to give a sense of the potential magnitude of iceberg melt and its influence?

*Authors' Response: We agree that given the temperate conditions of this fjord, iceberg melting would indeed be occurring. However, we have ignored iceberg melt in our estimations of*

*iceberg residence time because we do not have direct measurements on melt rates. Therefore the residence times we calculated would be a lower estimate of residence time. We have clarified this in the text. The difficulty with estimating iceberg melt in this situation is the lack of information about iceberg velocity and fjord water velocity. Because our images are taken too far apart, we are unable to track individual icebergs and thus calculate their rate of deterioration.*

8/1. The sentence says "evidence *for this*", but the sentence before mentions two possibilities. "This" must be specified.

*Authors' Response: We have added in a statement clarifying what is meant here. Thank you for catching this problem.*

9/12. What is the area of interest for "all of Columbia Fjord"? Please show on a figure.

*Authors' Response: What is meant by "all of Columbia Fjord" is the entire fjord as defined in the introduction. We have re-worded this sentence in the manuscript text to make this more clear.*

Section 4.3. This is a shallow discussion of iceberg influence on freshwater. As some points the authors mention temperature changes, at others they mention salinity. Both are likely affected, but this is not well discussed.

*Authors' Response: We recognize that this paragraph is too general and does not present significant results. We have re-worked this paragraph to focus more on results, and have created a new figure accompanying this paragraph comparing iceberg keel depths to salinity profiles in the fjord. In addition, we have made estimations of iceberg melt throughout the fjord.*

11/2. How can the authors conclude that winter-time capture of ice fragments aren't important when they did not examine winter-time data? The overall discussion in this paragraph also fails to recognize that iceberg residence time is only one factor in how icebergs change ocean water characteristics. For example, ocean water temperature, salinity, and current velocity and direction are also important.

*Authors' Response: We have imagery from March 13th, which is before the vernal equinox, i.e., the formal end of the astronomical winter. We do realize that it is standard in meteorological and climatic analyses to consider only December, January, and February to be winter months. However, the winter maximum of Arctic sea ice extent is typically reached in March, often around the time of when our March 13th scene was acquired. Hence, we do not think that it is too much of a stretch to consider this scene to be representative of winter time iceberg conditions in Columbia Fjord. We agree with the reviewer that this section does not address some important ways in which icebergs may affect fjord waters. We are working to address this.*

11/19. Why would the icebergs not be significantly rougher than the calving front?

*Authors' Response: Because both icebergs and the terminus are made of the same material and exposed to the same ocean water, we assumed that they would exhibit similar roughness. In our calculation of iceberg and glacier terminus surface area we made the simplifying assumption that they have the same roughness, given that we have no observational constraints on roughness for either surface. We will discuss this assumption in more detail in the context of current literature on this topic.*

12/4. What evidence is there to attribute the iceberg size change to increased crevasse hydro-fracture? Can the authors cite literature on this or point to observations that suggest this (even qualitatively)?

*Authors' Response:  Thank you for highlighting the problem with our discussions of this issue. Several previous studies have shown that power-law exponents of -1.5 are an indication of brittle fragmentation.  The summer power-law exponent values we find in this study are closer to -1.5 in the summer than in the spring or fall. This seasonal evolution of the power-law exponent supports the proposition that hydro fracture is an important process in summer, when surface melt rates are high and water-filled fractures should be abundant. We have improved our discussion of these issues in the revised manuscript in response to this comment of the reviewer.*

12/6. How can the authors identify a correlation between anything and average annual calving rate given that they have less than 1 year of data?

*Authors' Response:  Columbia Glacier velocity, flux, and terminus position data presented in reports from the US Geological Survey and Vijay and Braun (2017) indicate that 2013 was a reasonably representative year in terms of iceberg calving rates.  We therefore feel it is justified to compare our observations of icebergs in Columbia Fjord in 2013 to an average annual calving rate that was calculated for 2013.*

12/17. Alaska glaciers contributing to sea level rise is poor justification for studying icebergs in and of itself. I feel the authors need to think more deeply about what is unknown and the most important implications of their work.

*Authors' Response:  We thank the reviewer for this comment.  We have expanded our justification for studying Alaskan icebergs to include other reasons why icebergs may be important.*

Figure 1. It would be useful to label Heather Moraine in the image. 'PWS' label in c. is unnecessary. Is it possible to add the outline of the region in d. to the c. panel?

*Authors' Response:  We had previously outlined Heather Moraine in panel c, however, we have now added a label to make our outline more clear and noticeable to the reader.  We have removed PWS from panel c.  Instead of outlining the region covered by panel d in panel c, we have moved that bathymetry shown in panel d so that it overlaps the fjord extent shown in panel c.*

Figures 4 and 8. It would be nice to put this data on a map/image background.

*Authors' Response:  We have added in a satellite image of the fjord taken on June 10, 2013 to the background of figures 4 and 8 to give the reader further spatial context.*

Figure 5. Why not show data from all periods? Also, it would be preferable to list the class sizes in the captions.

*Authors' Response:  We agree that it would be preferable to show the data from all dates.  We have included plots from the other images.  In addition, we have listed the class sizes in the figure caption.*

Figure 6. Bad data should not be included in the plot.

*Authors' Response:  We assume the reviewer is referring to the November 19a data.  We have removed this data from the plot entirely.*

Table 2. What is the source of these errors? What does this error represent?

*Authors' Response: The errors are the upper and lower limits of the 95% confidence interval.*

Tables 3 and 4. Again, bad data should not be included in plots or tables. Either fix it or remove it.

*Authors' Response: We assume the reviewer is referring to the data from November 19a. We have removed this data from both tables.*

**Technical corrections (by page/line number):**

2/20. Please remove "in order" for all cases of "in order to". It is unnecessary.

*Authors' Response: We have changed all cases of "in order to" to simply "to."*

---

## Author Comment (AC4) · 14 Mar 2019

Review of tc-2018-230 Spatiotemporal Distributions of Icebergs in a Temperate Fjord: Columbia Fjord, Alaska by S. Neuhaus et al

General comment

The paper analyzes the distribution of icebergs within the Columbia fjord using high resolution (0.5 m) visible satellite images (10 covering a 8 months period in 2013). Classical image processing tools (from Matlab) are then used to detect and estimate the size of the icebergs within the fjord. The results are validated by comparison with manual analysis for selected scenes. The results of the images analysis are then used to compute the time/space distribution of iceberg and the evolution of the distribution of the icebergs' size in time and location within the fjord. The analysis reveals the complex distributions of the iceberg within the fjord as well as the seasonal variability related to the glacier calving rate.

The results are of interest and are worth publishing. However, the motivation of the study should be stated more clearly and if the implication for the future of Greenland icebergs (if kept) should be better justified.

The study is also limited to 8 month and 10 images where certainly much more are available. I know that image processing is quite hard and fastidious but at least the authors should justified why they limited their study to this short data set. Paragraph 4.3 and 4.4 need to be better focused on real results and not on quite shallow general discussions. The distribution of volume and the evolution of the size distribution are important results by themselves. I think that the study could also be improved if simple computation of freshwater flux using ice volume and classical melting law were conducted and presented.

*Authors' Response: We would like to thank the reviewer for their comments.*

*We would like to thank the reviewer for drawing to our attention the fact that our discussion of Columbia Fjord as a potential future Greenlandic fjord is too general.  We have made sure to address this in our revisions.  Instead of justifying our study by presenting Columbia Fjord as a potential future Greenland fjord, we have focused our manuscript on the more specific implications of our results.  We have amended section 4.4 so that it is now a more detailed comparison of our findings to measurements which have been previously published in papers covering the topic of icebergs in fjords.*

*We have also worked on section 4.3 to make it less general and more focused on our results. We have done this by adding estimated iceberg melt from melt equations published in Bigg et al. (1997), as well as presenting salinity profiles taken inside Columbia Fjord during our study period by the US Geological Survey.*

Specific comments
Page 1 line 8 outet=> outlet

*Authors' Response:  We appreciate the reviewer catching this typo.  We have corrected it.*

- Page 1 line 20: Coloumbia==> Columbia

*Authors' Response: We appreciate that the reviewer caught this typo as well.  We have corrected it.*

- Page 1 line 20 Considering the difference of temperature between Greenland and PWS water and the different conditions of the Greenland fjords this remark is certainly way to general.

*Authors' Response: We recognize that our comparison of Prince William Sound in Alaska and fjords in Greenland was far too general.  We are currently working on correcting this.*

- Page 3 lines 15-20. For people not familiar with WorldView Satellite explain why there are sometimes two images from the same satellite at the same time. The sentence on the hundredths of second separation is quite useless.

*Authors' Response: We agree that there should be an explanation of why the WorldView Satellite would take two images only minutes apart.  The images were taken for the purposes of DEM creation using stereo imagery.  We have thus added in a few sentences in Section 2.1 explaining this.  We have also removed the sentence mentioning "hundredths of a second" because it is unhelpful.*

- Page 3 line 23 and following; As the study fully relies on the detection and analysis of the WW1 images, it is important to at least present an example of detection (on an image detail) at best to provide the analysis of all images in Supplementary Information.

*Authors' Response:  To give readers a better idea of what we have done, we have added in an example of iceberg detection to figure 2, as well as adding in a sentence referencing said figure.*

- Page 4 lines 23-26. It is not explain what is the difference between a and b images ( see my previous comment). If this is related to different viewing angle it is important to precise it as it might explain the different result (that could be due to a difference in ef- fective resolution). I don't understand the November 19 case (not enough information). There again it could be related to viewing angle and specular reflection on open water (wild guess as we don't have the data and there are not freely accessible).

*Authors' Response:  Because on a few instances there were two sets of stereo images taken, we had two images taken on the same day.  We labelled these images 'a' and 'b' respectively.  We have now added in a few sentences distinguishing 'a' and 'b' images to the text, as well as mentioning that the difference in iceberg identification could be due to viewing angle.  Changing the angle at which we view the ocean changes the reflectivity of the ocean, which could therefore affect which pixels were identified as ice versus water.*

- Page 4 line 32. Why May 06 (a b) is not included? Provide explanation. reference to figure 4 should be included.

*Authors' Response:  Omitting May 06a from the figure was a mistake on our part.  We have fixed this by adding the icebergs from May 06a into figures 4a and 4b.  We have also added references in the text to these two figures.*

- Page 5 line 13-14, The sentence is not very clear. The pdf is computed on the following bins.

*Authors' Response:  We have re-worded this sentence to make it less confusing.*

- Page 5 &2.4. This paragraph presents two methods of estimation of the iceberg volume from the satellite iceberg area (which might be different from the waterline cross- sectional area if the water is very clear). A is not a proxy.

*Authors' Response: We do not use waterline cross-sectional area as a proxy for iceberg volume as we actually calculate iceberg volume using two different methods.  We have removed the mention of using waterline cross-sectional area as a proxy.*

- Page 6 line 7-. I think the authors recompute the albedo using fixed ocean and ice value to eliminate solar angle and atmospheric influence but it is not stated.

*Authors' Response:  Yes, we calculated the albedo using this method in order to avoid issues with atmospheric interference as well as the angle at which solar radiation hits the ocean.  We have added in a sentence stating this more clearly.*

- Page 6 lin18-20 Where is the 95% coming from. Figure 4 and 5 don't present proportion but numbers and area. Figure 4 should use a log colorscale to reveal more details of the distribution within the fjord.

*Authors' Response:  95% of all the icebergs identified in this study had a waterline cross-sectional area less than or equal to 100 $m^2$.  As this is not readily discernible from the figure, we are amending it, as well as re-wording the text to make this more clear. We have also changed the colorscale of figure 4 to be logscale in order to emphasize iceberg distributions in the fjord.*

- Page 6 line 20-24. I don't understand the purpose of this remark. It is part of the detection and analysis method and should be treated there.

*Authors' Response:  We have moved this to the methods section.*

- Page 6 line 25. Please mention figure 6 from the beginning of the &.

*Authors' Response: We have added in a reference to figure 6 in the first sentence of the paragraph.*

- Figure 6. For May 6 b and Jul 11 there are only estimates for the proximal zone certainly because of the partial coverage. Is it really necessary to keep those two images as they don't really bring any special information/results.

*Authors' Response: Despite the fact that there is incomplete coverage for the fjord on May 06b and July 11, we believe there is value in representing the partial data in the figure.  The power-law exponents for these two dates show agreement with the power-law exponents in the proximal zone for the other dates in the same seasons.*

- Page 7 line 13-15. The comparison of the thickness for a given A using 2 and 3 explains completely the difference observed in Table 3. What is important to note here is that the variations of the total volume, and percentage of volume for large icebergs are very similar using the two formulas although they give very different iceberg thickness (this is certainly due to the strong impact of the power-law distribution of the distribution of the ice volume).

*Authors' Response: This is a very good point.  We are now addressing this in the text.*

- Page 7 line 19. The summer increase of albedo could implies that the fragmentation is increasing in warmer waters.

*Authors' Response:  This is an interesting point.  We have added a sentence addressing this into the discussion section.*

- Page 7 line 21-25. It is important to explain in detail the computation of the residence time.

*Authors' Response: We have clarified our computation of iceberg residence time by stating our methods more explicitly.*

- Page 8 line 23-25 In fact power-law and lognormal are quite similar and power-laws (which do not converge (tend to infinity in 0) ) can be used to approximated the tail of lognormal distributions.

*Authors' Response:  This is an interesting point, and something we are working to address in the manuscript.*

- Page 9 line 7-17 Two recent studies one from Bouhier et al (https://doi.org/10.5194/tc-12-2267-2018) and one from Crawford et al (https://doi.org/10.1029/2018JC014388) presented size distributions of pieces resulting from icebergs fragmentation with slope close to -1.5 (i.e. the mid_fjord summer slope). The two studies mentioned that this -1.5 slope is in general associated to fragile fragmentation and could in this case indicate that during summer month the main driver of the size distribution within the fjord is the fragmentation.

*Authors' Response: We very much appreciate the reviewer bringing these two studies to our attention. We have used their findings, as well as those from other studies of iceberg fragmentation, to expand on brittle fragmentation. We agree that the power-law exponents we find in this study indicate that there is more brittle fragmentation of icebergs during the summer months than the spring or fall. We have added in a number of sentences to the discussion on this topic.*

- Page 9 line 26-27 In fact when computing power-law distribution there is always some problem with the tail of the distribution just because the numbers of samples is too low.

*Authors' Response: The reviewer makes a very good point. We have added in a sentence explaining this in the methods section where we first mention removing the tail of the distribution to achieve a better fit.*

Page 9 line 30. This is a direct consequence of the power law distribution.

*Authors' Response: We agree with this point, and have added in a statement pointing this out in the manuscript text.*

- Page 9-10 &4.3 This paragraph is not very clear and don't present any significant results. It could be of interest if the volume of ice and melting law were used to estimate the fresh water flux from each image.

*Authors' Response: We recognize that this paragraph is too general and does not present significant results. We have re-worked this paragraph to focus more on results, and have created a new figure accompanying this paragraph comparing iceberg keel depths to salinity profiles in the fjord to visually represent the contribution of freshwater from icebergs at various points along the fjord length. In addition, we have made estimations of iceberg melt using an equation from Bigg et al. (1997), which we were able to use by making assumptions about fjord conditions and iceberg velocity.*

- Page 10 &4.4 There is no data in winter in your study.

*Authors' Response: In the arctic, the sea ice maximum extent is reached in March. Because there is no sea ice present in our March imagery, we made the assumption that there was no significant sea ice present in the fjord.*

- Page 11 line 1 Where is 11% mélange coming from (not from Table4).

*Authors' Response: We calculated the mélange by subtracting the total number of ice pixels by the number of ice pixels that are identified as part of an iceberg. We have included our calculations of mélange into the methods section.*

- Page 11 line 14-15. The computation of the iceberg surface is not obvious. It should be given (in annex).

*Authors' Response: We agree that we do not explicitly explain our calculations of iceberg surface area. We are adding the explanations of iceberg surface area calculations into the methods.*

- Page 12 line 4-6. The -1.5 slope could indicate that there is more fragmentation during summer.

*Authors' Response: We agree with the reviewer on this point. We have therefore edited this sentence to include brittle fragmentation.*

---

## Author Response (AR1)

[revised manuscript text omitted]

| λ  | Deleted: the                                                             |
|----|--------------------------------------------------------------------------|
| 1  | Deleted: National Oceanographic and Atmospheric
Administration (NOAA) |
| ~( | Deleted: )                                                               |
| -( | Deleted: After the initial increase in                                   |
| (  | Deleted: i                                                               |
| -( | Deleted: -                                                               |
| (  | Deleted: c                                                               |

[revised manuscript text omitted]
     |                                                                                                                                                                     |  |  |  |  |  |
| Deleted: 15 ± 5 days using Eq. (3)                 |                                                                                                                                                                     |  |  |  |  |  |
| Deleted:
represent th
we do not a            | $13 \pm 6$ days when using Eq. (2). These estimates

[revised manuscript text omitted]

---

## Author Response (AR2)

Reviewer 1: There remain a few small technical errors that should be addressed with a detailed read through by the authors, but other reviewer comments have been addressed appropriately.

*Thank you for your comments.  We have done a careful read-through of the manuscript and fixed a few spelling errors as well as tightening up some of the wording.*

Reviewer 2: I think this is a very interesting study of the icebergs calving from a glacier in a rapidly changing environment. It is important to have a description of the present state and distribution of icebergs within the fjord.
My general and specific comments made on the original version have been taken into account and the revised manuscript has gain in strength and significance (melt water analysis, T-S profiles.).

*Thank you for your comments.*

[revised manuscript text omitted]